# LOW-COST ALGORITHMIC RECOURSE FOR USERS WITH UNCERTAIN COST FUNCTIONS

## ABSTRACT

The problem of identifying algorithmic recourse for people affected by machine learning model decisions has received much attention recently. Existing approaches for recourse generation obtain solutions using properties like diversity, proximity, sparsity, and validity. Yet, these objectives are only heuristics for what we truly care about, which is whether a user is satisfied with the recourses offered to them. Some recent works try to model user-incurred cost, which is more directly linked to user satisfaction. But they assume a single global cost function that is shared across all users. This is an unrealistic assumption when users have dissimilar preferences about their willingness to act upon a feature and different costs associated with changing that feature. In this work, we formalize the notion of user-specific cost functions and introduce a new method for identifying actionable recourses for users. By default, we assume that users' cost functions are hidden from the recourse method, though our framework allows users to partially or completely specify their preferences or cost function. We propose an objective function, *Expected Minimum Cost* (EMC), based on two key ideas: (1) when presenting a set of options to a user, it is vital that there is at least one low-cost solution the user could adopt; (2) when we do not know the user's true cost function, we can approximately optimize for user satisfaction by first sampling plausible cost functions, then finding a set that achieves a good cost for the user in expectation. We optimize EMC with a novel discrete optimization algorithm, *Cost-Optimized Local Search* (COLS), which is guaranteed to improve the recourse set quality over iterations. Experimental evaluation on popular real-world datasets with simulated user costs demonstrates that our method satisfies up to 25.89 percentage points more users compared to strong baseline methods. Using standard fairness metrics, we also show that our method can provide more fair solutions across demographic groups than comparable methods, and we verify that our method is robust to misspecification of the cost function distribution.

## 1 INTRODUCTION

Over the past few years ML models have been increasingly deployed to make critical decisions related to loan approval (Siddiqi, 2012), insurance (Scism, 2019), allocation of public resources (Chouldechova et al., 2018; Shroff, 2017) and hiring decisions (Ajunwa et al., 2016). Decisions from these models have real-life consequences for the individuals (users) involved. As a result, there is a growing emphasis on explaining these models' decisions (Ribeiro et al., 2018; Lundberg & Lee, 2017; Poulin et al., 2006) and providing *recourse* for unfavorable decisions (Voigt & dem Bussche, 2018; Karimi et al., 2020a). A *recourse* is an actionable plan that is given to someone allowing them to change the decision of a deployed model to a desired alternative (Wachter et al., 2017). Recourses can be highly valuable for users in situations where model decisions determine important life outcomes. Or, in cases where no feasible recourse is possible, users may wish to dispute the use of a model in the first place, and we might take this as evidence that greater reforms to the decision-making system are needed (Venkatasubramanian & Alfano, 2020).

Recourses are desired to be *actionable*, *feasible*, and *non-discriminatory*. *Actionable* means that only features which can be changed by the user are requested to be changed. These changes should also be possible under the data distribution. For example, *Education level* cannot be decreased from a *Masters* to *Bachelors* degree but can be increased from *Masters* to *PhD*. It is also not actionable to change your *Race* (Mothilal et al., 2020). A recourse is *feasible* if it is reasonably easy for the

user to adopt, i.e. it is actionable and has a low cost for the user. *Non-Discriminatory* means that the recourse method should be equitable across population subgroups. There are now many fairness metrics that can be used to measure this (Hinnefeld et al., 2018), e.g. the ratio between the average cost of recourse for two subgroups in a population.

While we want recourses to be feasible for all users, it is difficult to directly optimize for a user's incurred cost unless we have access to their ground-truth cost function. In the absence of detailed cost function data, prior work has used other heuristic objectives for feasibility. For instance, Mothilal et al. (2020) and Wachter et al. (2017) assume that if the vector distance between the user's current state and the recourse is small, then recourse will be low cost. These works encourage this property via a *proximity* objective. Meanwhile, *sparsity* quantifies the number of features that require modification to implement a recourse (Mothilal et al., 2020). When providing multiple recourse options, *diversity* in proposed recourses is used to counter uncertainty around the user cost function (Mothilal et al., 2020; Cheng et al., 2021). The assumption is that if users are provided with diverse options then they are more likely to find at least one feasible solution. Later in section 5.2, we show that diversity as an objective correlates poorly with user cost, suggesting it would be strongly preferable to optimize for user cost directly.

A few approaches to recourse do directly optimize for the cost of recourse, but they assume there is a single cost function shared by all users (Ustun et al., 2019; Rawal & Lakkaraju, 2020; Karimi et al., 2020c;d; Cui et al., 2015). We believe it is crucial to have user-specific cost functions, as a global cost function might poorly represent different users in a diverse population.

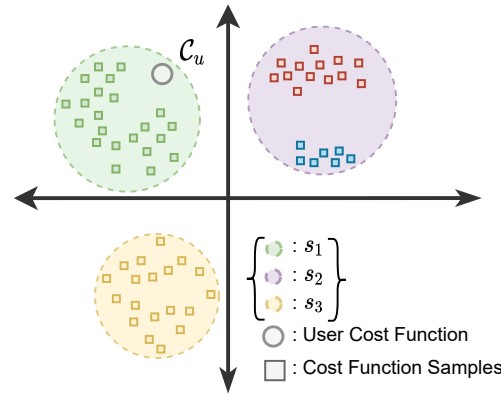

Figure 1: Method diagram showing the intuition behind the Expected Minimum Cost Objective. The squares denote cost function samples, which are the same color when they are similar. We aim to find a solution set of generated counterfactuals where each counterfactual does well under a particular region of cost function space (here, $\{s_1, s_2, s_3\}$). The shaded regions each represent a set of cost functions which a single $s_i$ caters to. In this case, we do not have enough counterfactuals to "cover" every region of the cost function space, so a single counterfactual ($s_2$) must cater to two different regions. Here the user's hidden ground-truth cost function, $\mathcal{C_u}$, is served well by $s_1$.

In this work, we propose a method for identifying a user-specific recourse set that contain at least one good solution for the user. We directly optimize for user cost by quantifying the actionability and feasibility of proposed recourses. In the absence of data about user cost functions, we treat them as hidden from the recourse method and assume they follow an underlying cost distribution. However, we provide users with an option to specify their preferred editable features or the complete cost function detailing the costs of transitions between features values. We model this cost distribution via a highly flexible hierarchical cost sampling procedure which makes minimal assumptions about user preferences (Algorithm 1). Based on this distribution, we propose an objective function, *Expected Minimum Cost* (EMC), which allows us to approximately optimize for user satisfaction by first sampling plausible cost functions, then finding a set that achieves a good cost in expectation. The EMC objective encourages the solution set to consist of counterfactuals that are each a good counterfactual under some particular cluster of cost functions from the distribution. Hence, no matter what the user's ground-truth cost function is, we will have *some* counterfactual that is well suited to the user's cost function (shown in Figure 1). Next, we propose a simple discrete optimization method, *Cost-Optimized Local Search* (COLS), in order to optimize for Expected Minimum Cost. COLS guarantees a monotonic reduction in the Expected Minimum Cost of the counterfactual set, which we show leads to large empirical reductions in user-incurred cost.

To evaluate the effectiveness of our proposed techniques, we run experiments on two popular real-world datasets: Adult-Income (Dua & Graff, 2017) and COMPAS (Larson et al., 2016). We compare our method with multiple strong baselines methods like Diverse Counterfactual Explanations (DICE) (Mothilal et al., 2020), Feasible and Actionable Counterfactual Explanations (FACE) (Poyi-

adzi et al., 2020), and Actionable Recourse (AR) (Ustun et al., 2019). We evaluate these methods on existing metrics from the literature like diversity, proximity, sparsity, and validity (Section 5.1) along with two cost-based metrics. In particular, we measure the *fraction of satisfied users*, based on whether their cost of recourse is below a certain threshold $k$. We also report *coverage*, which is the fraction of users with at least one actionable recourse (Rawal & Lakkaraju, 2020). Using simulated user cost functions, we show that our method satisfies 25.89% and 17.93% percentage points more users than strong baseline methods while covering 22.35% and 17.13% more users, on the Adult-Income and COMPAS dataset respectively. We perform important ablations to show whether performance can be attributed to the COLS optimization method or the EMC objective. Additionally, we evaluate the robustness of our method to various distribution shifts that can occur between the user's hidden cost distribution and the hierarchical cost sampling distribution. We find that our method is robust to these distribution shifts and generalizes well to user cost function from these shifted distributions. Lastly, we perform a fairness analysis of all the methods across demographic subgroups based on *Gender* and *Race*. Standard fairness metrics demonstrate that, in most comparisons, our method is more fair relative to the strongest baseline methods.

Our primary contributions in this paper are listed below.

1. We propose to evaluate user-incurred cost and fraction of satisfied users by means of hidden user-specific cost functions, rather than a known global cost function.
2. We propose a new objective function, Expected Minimum Cost (EMC), which allows us to approximately optimize for user satisfaction when user cost functions are not known.
3. We propose a discrete optimization method, *Cost-Optimized Local Search* (COLS), which achieves up to 25.89% percentage points higher user satisfaction relative to the next-best baseline. COLS guarantees a monotonic reduction in EMC, which we find provides a 19% point improvement over a simple local search.
4. We show that in most settings, COLS provides more fair solutions across demographics subgroup than comparable recourse methods, while offering recourse to a substantially higher fraction of users.

## 2    RELATED WORK

A wide variety of methods have been proposed for generating recourses. For a comprehensive survey of existing recourse methods, we refer readers to Karimi et al. (2020b). Here, we distinguish our approach based on our recourse objectives, optimizer, and evaluation. We primarily discuss recourse methods, though there is useful complementary work on problems such as providing recourse when there is distribution shift in the data (Upadhyay et al., 2021; Slack et al., 2021) and training models which guarantee recourse to affected individuals with high probability (Ross et al., 2021).

**Objectives:** The most prominent family of objectives for recourse includes distance-based objectives (Wachter et al., 2017; Karimi et al., 2020a; Dhurandhar et al., 2018; Mothilal et al., 2020; Rasouli & Yu, 2021). These methods seek recourses that are close to the original data point. In DICE, Mothilal et al. (2020) provide users with a set of counterfactuals while trading off between proximity, a distance-based objective, and diversity. Diversity-based methods assume that providing diverse options will increase the chance a user is satisfied by one of the options. A second category of methods uses other heuristics based on the data distribution (Aguilar-Palacios et al., 2020; Gomez et al., 2020) to come up with counterfactuals. FACE constructs a graph from the given data and then tries to find a high-density path between points in order to generate counterfactuals (Poyiadzi et al., 2020). Lastly, the works closest to ours are the cost-based objectives, which capture feasibility in terms of the cost of recourse: (1) Cui et al. (2015) define a cost function based on the minimum and maximum values a factor can take in their additive tree model. (2) Karimi et al. (2020c;d) take a causal intervention perspective on the task and define cost in terms of the normalized distance between the user state and the counterfactual. (3) Ustun et al. (2019) define cost in terms of the number of changed features and frame recourse generation as an Integer Linear Program. (4) Rawal & Lakkaraju (2020) infer a cost function from simulated rankings of features for actionability, then optimize recourses for this cost function. Importantly, all of these works assume there is a known global cost function that is shared by all users. In our work, we drop this assumption, and instead we optimize for cost over a distribution of plausible user-specific cost functions.

**Optimization:** Early work on recourse methods uses gradient-based optimization to search for counterfactuals close to a user's data point (Wachter et al., 2017). Several methods since then also use gradient-based optimization (Mothilal et al., 2020; Chen et al., 2020). Some recent approaches

use tree-based techniques (Rawal & Lakkaraju, 2020; Aguilar-Palacios et al., 2020; von Kügelgen et al., 2020; Hashemi & Fathi, 2020; Kanamori et al., 2020), kernel-based methods (Dandl et al., 2020; Gomez et al., 2020; Ramon et al., 2020), while others employ some heuristic (Poyiadzi et al., 2020; Aguilar-Palacios et al., 2020) to generate counterfactuals. A few works use latent space perturbation with autoencoders to generate recourses (Pawelczyk et al., 2020; Joshi et al., 2019), while Karimi et al. (2020a) and Ustun et al. (2019) utilize SAT and ILP solvers, respectively. Here, we introduce a discrete optimization method specialized for our Expected Minimum Cost objective.

**Evaluation:** Besides ensuring that recourses are classified as the favorable class by a model (validity), the most prominent approaches to evaluate recourses rely on Distance-based metrics. In DICE, Mothilal et al. (2020) evaluate recourses according to their proximity, sparsity, and diversity. Meanwhile, several works directly consider the cost of the recourses, using a known global cost function as a metric, meaning that all users share a cost function, which is available to the recourse generation method (Cui et al., 2015; Karimi et al., 2020c;d). In a slight departure from this setting, Rawal & Lakkaraju (2020) estimate a cost function from simulated pairwise feature comparisons, but this single estimate is used for both recourse generation and evaluation. In contrast, we evaluate a recourse method by simulating user-specific cost functions which can very greatly across users, and these cost functions are not known in advance to recourse generation methods. We will also measure recourse coverage as defined by Rawal & Lakkaraju (2020), which measures the fraction of users that were provided with a recourse by the method.

## 3 PROBLEM STATEMENT

**Notation.** We assume that we have a dataset with features $\mathcal{F} = \{f_1, f_2, ... f_k\}$. Each feature can either be continuous $\mathcal{F}^{con} \subset \mathcal{F}$ or categorical $\mathcal{F}^{cat} \subset \mathcal{F}$. Each continuous feature $f_i^{con}$ takes values in the range $[r_i^{min}, r_i^{max}]$, which we discretize to integer values. For a continuous feature $f_i$, we define the range $Q^{(f_i)} = \{k \in \mathbb{Z} : k \in [r_i^{min}, r_i^{max}]\}$ and for a categorical feature $f_i$, we define it as $Q^{(f_i)} = \{q_1^{f_i}, q_2^{f_i}, ..., q_{d_i}^{f_i}\}$, where $q_{(.)}^{f_i}$ are the states that feature $f_i$ can take. Features can either be mutable ($\mathcal{F}^m$), conditionally mutable ($\mathcal{F}^{cm}$), or immutable ($\mathcal{F}^{\oslash}$), according to the real-world causal processes that generate the data. Mutable features can transition from between any pair of states in $Q^{(f_i)}$; conditionally mutable features can transition between pairs of states only when permitted by certain conditions; and immutable features cannot be changed under any circumstances. For example, *Race* is an immutable feature (Mothilal et al., 2020), *Age* and *Education* are conditionally mutable (cannot be decreased under any circumstances), and *number of work hours* is mutable (can both increase and decrease). Lastly, while continuous features inherently define an ordering in its values, categorical features can either be ordered or unordered based on its semantic meaning. For instance, *Age* is an ordered feature that is conditionally mutable (can only increase).

**Cost Function.** In this work, we assume that each user has a hidden preference regarding the ease of changing a particular feature, where different users can have different preferences. Such differential preferences can be expressed via user-specific costs of transitioning between feature states. We define a *cost function* for each user $\mathcal{C} : \mathbb{R}^{|\mathcal{F}|} \to \mathbb{R}^{|\mathcal{F}|}$ as a set of feature-specific functions which provide the user-incurred cost when transitioning between feature states. Formally, a cost function is parametrized as a set of *transition matrices*, $\mathcal{C} = \{\mathcal{C}^{(f)} \in \mathbb{R}^{|Q^{(f)}| \times |Q^{(f)}|} \mid \forall f \in \mathcal{F}\}$, where each element of the transition matrix $\mathcal{C}_{ij}^{(f)} \in [0, 1] \cup \{\infty\}$ is the cost of transition from state $i \to j$ for the feature $f$. Here, $0$ means the transition has no associated cost incurred, and $1$ means the transition is maximally difficult to make. Infeasible transitions have a cost of $\infty$ (associated with immutable features). For the remainder of the paper, we interchangeably write $\mathcal{C}^{(f)}$ as matrix or a function where $\mathcal{C}^{(f)}(m, n) = \mathcal{C}_{m,n}^{(f)}$.

**User Definition.** A user $\boldsymbol{u}$ is defined as a tuple $\boldsymbol{u} = (\boldsymbol{s}_u, \mathcal{C}_u^*)$, where $\boldsymbol{s}_u$ is the state vector of length $|\mathcal{F}|$ containing the user's features values and $\mathcal{C}_u^*$ is their unknown ground-truth cost function (see Appendix Table 5 for qualitative example). This ground-truth cost function, $\mathcal{C}_u^* = \{\mathcal{C}_u^{*(f)} \mid \forall f \in \mathcal{F}\}$, provides the cost of transitioning between different feature states. Following past work (Rawal & Lakkaraju, 2020), we note that it may be difficult for users to precisely quantify their own cost functions in practice, but we do assume that each user has some hidden cost function that specifies which recourses they would prefer. Different from past work (Rawal & Lakkaraju, 2020; Ustun et al., 2019), we suppose that each user in a population has their own cost function, rather than assuming there is a global cost function (single shared and fixed cost function across users).

**Transition Costs.** Given two state vectors $\boldsymbol{s}_i$, $\boldsymbol{s}_j$ and any cost function $\mathcal{C}$, transition cost is the summation of transition costs for individual features, defined as $\text{Cost}(\boldsymbol{s}_i, \boldsymbol{s}_j; \mathcal{C}) = \sum_{f \in \mathcal{F}} \mathcal{C}^{(f)}(s_i^{(f)}, s_j^{(f)})$, where $\boldsymbol{s}^{(f)}$ is the value of feature $f$ in the state vector. Now, given a recourse set $\mathcal{S}$, we suppose that the cost a user will incur is the *minimum* transition cost across possible recourses, since a rational user will select the least costly option. So, given a user $\boldsymbol{u}$ with state vector $\boldsymbol{s}_u$ and a counterfactual set $\mathcal{S}$, the cost of transition under a cost function $\mathcal{C}$ is defined as

$$\text{MinCost}(\boldsymbol{s}_u, \mathcal{S}; \mathcal{C}) = \min_{s_j \in \mathcal{S}} \text{Cost}(\boldsymbol{s}_u, \boldsymbol{s}_j; \mathcal{C}). \tag{1}$$

**Problem Definition.** For a given user $\boldsymbol{u}$ our goal is to find a recourse set $\mathcal{S}_u$ such that,

$$\mathcal{S}_u = \arg\min_{\mathcal{S}} \text{ MinCost}(\boldsymbol{s}_u, \mathcal{S}; \mathcal{C}_u^*) \text{ s.t. } \exists\, s_i \in \mathcal{S} \text{ s.t. } f(s_i) = +1 \tag{2}$$

where $f$ is the black-box ML model and $+1$ is the desired outcome class. We want to offer users at least one counterfactual that is a good solution under their hidden cost function $\mathcal{C}_u^*$. Since we are uncertain about which cost function is the user's true cost function, we minimize the expected cost for each user using a distribution over plausible cost functions (Section 4.1).

**Measuring Recourse Quality.** While we could measure average population cost as our metric for recourse method quality, it is also helpful to ask: what proportion of users are *satisfied* with the recourses they are given? We say that a user is *satisfied* by a recourse set if the best option in that set achieves a sufficiently low cost. Formally, given a set of users $\mathcal{U}$ and a set of sets of generated counterfactuals $\{\mathcal{S}_u\}_{u \in \mathcal{U}}$ from a recourse method, the fraction of user's satisfied at a particular cost threshold $k$, FS@k, can be defined as:

$$FS@k(\mathcal{U}, \{\mathcal{S}_u\}_{u \in \mathcal{U}}) = \sum_{u \in \mathcal{U}} \frac{\mathbb{1}_{\{\text{MinCost}(\boldsymbol{s}_u, \mathcal{S}_u; \mathcal{C}_u^*) < k\}}}{|\mathcal{U}|} \tag{3}$$

We note that optimizing Equation 2 is equivalent to finding the set of counterfactuals that maximizes the probability that a user is *satisfied* by some counterfactual in the set.

Another important quantity is the *Coverage (Cov)* which measures the fraction of users to which the recourse algorithm can provide any actionable recourse.

$$Cov(\mathcal{U}, \{\mathcal{S}_u\}_{u \in \mathcal{U}}) = \sum_{u \in \mathcal{U}} \frac{\mathbb{1}_{\{\text{MinCost}(\boldsymbol{s}_u, \mathcal{S}_u; \mathcal{C}_u^*) < \infty\}}}{|\mathcal{U}|} \tag{4}$$

## 4 PROPOSED METHOD: EMC AND COLS

In this section, we define our proposed objective function, *Expected Minimum Cost* (EMC), cost sampling distribution, and *Cost-Optimized Local Search* (COLS) optimization method.

### 4.1 APPROXIMATELY OPTIMIZING FOR USER COST UNDER UNCERTAIN COST FUNCTIONS

In most use cases, the cost functions associated with each user $\mathcal{C}_u^* \sim \mathcal{D}^*$ are unknown to us and follow the population's true cost function distribution $\mathcal{D}^*$ which is also unknown. Hence, we cannot exactly minimize the user cost in Equation 1 with respect to $\mathcal{C}_u^*$. Yet we can approximately optimize for the true user cost $\mathcal{C}_u^*$ by means of a plausible cost function sampling distribution $\mathcal{D}_{train}$ if it can model diversity in user preferences and cost function effectively. Given samples from this distribution, we can minimize the *Expected Minimum Cost* (EMC) of a transition for a user. Formally, for any given user $\boldsymbol{u} = (\boldsymbol{s}_u, \mathcal{C}_u^*)$, we want to optimize for

$$\mathbb{E}_{\mathcal{C}_i \sim \mathcal{D}_{train}}[\text{MinCost}(\boldsymbol{s}_u, \mathcal{S}; \mathcal{C}_i)] \tag{5}$$

In practice, we employ Monte Carlo Estimation (Robert & Casella, 2010) to approximate this expectation by sampling $M$ cost functions $\{\mathcal{C}_i\}_{i=1}^M$ from $\mathcal{D}_{train}$, which we term our EMC objective:

$$\text{ExpMinCost}(\boldsymbol{s}_u, \mathcal{S}; \{\mathcal{C}_i\}_{i=1}^M) = \frac{\sum_{i=1}^M \text{MinCost}(\boldsymbol{s}_u, \mathcal{S}; \mathcal{C}_i)}{M} = \frac{\sum_{i=1}^M \min_{\boldsymbol{s}_j \in \mathcal{S}} \text{Cost}(\boldsymbol{s}_u, \boldsymbol{s}_j; \mathcal{C}_i)}{M} \tag{6}$$

**Remark:** Note that there is an important interaction between the expectation and the MinCost terms in this objective. EMC encourages the solution set to consist of recourses that are each a good option under some particular *cluster* of cost functions from the distribution. That is, the $\min$ term allows for different recourses to cater to different regions of the cost function distribution. As a result, no matter what the user's ground truth cost function is, we are likely to have found at least one counterfactual that is well suited to that cost function (see Figure 1).

### 4.2 HIERARCHICAL COST SAMPLING PROCEDURE

To optimize for EMC, we need a plausible distribution $\mathcal{D}_{train}$ which is capable of generating cost functions which model a diverse set of user preferences. We propose, three hierarchical cost sampling distribution, $\mathcal{D}_{perc}$, $\mathcal{D}_{lin}$, $\mathcal{D}_{mix}$ which are highly flexible and makes minimal assumptions about the user preference (see Algorithm 1). The samples from $\mathcal{D}_{perc}$ and $\mathcal{D}_{lin}$ model the transition cost with percentile shift and linear cost respectively. Whereas, the samples from $\mathcal{D}_{mix}$ distribution models a linear combination of *percentile shift cost* (Ustun et al., 2019) and *linear cost*, where the mixture weights of this combination are user-specific, capturing their disposition towards these two types of costs. *Percentile shift cost* for ordered features is proportional to the change in a feature's percentile value with the change from an old feature value to a new one (see Algorithm 4). The *Linear cost* for ordered features is proportional to the number of intermediate states in $Q^{(f)}$ which a user will have to go through while transitioning from their current state to the final state (see Algorithm 5). For a detailed description of the sampling procedure, please refer to Appendix A.2.1 and Algorithm 1.

We emphasize a few core properties of $\mathcal{D}_{mix}$ distribution: (1) The distribution is very flexible. It is able to capture all possible feature subsets which different users might consider editable as well as possible mixing weights for combining linear and percentile costs. Critically, the difficulty of editing each feature can range from "trivial" to "maximally difficult" (relative to other features), meaning almost all plausible user cost functions should be represented in the distribution. (2) The cost distribution mean follows the monotonicity property, i.e. if the user has to make more drastic changes to the feature, then the associated cost will be higher. (3) The user has an option to adapt this distribution to their needs by providing either the editable features or the feature scores. Together, these properties allow us to represent a large space of plausible user cost functions, which helps us ensure our recourses can satisfy a diverse user preferences.

### 4.3 SEARCH METHODS FOR FINDING LOW COST COUNTERFACTUALS.

**COLS:** To optimize for Expected Minimum Cost (Equation 6), we propose two simple, efficient, and optimized discrete search algorithms (Ebrahimi et al., 2018; Baptista & Poloczek, 2018), namely *Cost-Optimized Local Search* (COLS) and *Parallel Cost-Optimized Local Search* (P-COLS) (refer to Algorithm 2). COLS maintains a best set which will be the final recourse provided to the user. At each iteration, a candidate set is generated by locally perturbing each element of the best set with a Hamming distance of two, then it is evaluated against the EMC objective. Instead of making a direct comparison between the best-set-so-far and the candidate set, at this point we evaluate whether any counterfactuals from the candidate set would improve the best set if we swapped out individual counterfactuals. Specifically, if the benefit of replacing $s_i \in \mathcal{S}_t$ with $s_j \in \mathcal{S}^{best}$ is positive then we make the replacement. The ability to assess the benefit of each candidate counterfactual is critical because it allows us to constantly update the best set instead of waiting for an entire candidate set with lower EMC. For objectives like diversity, evaluating the benefit of individual replacement becomes prohibitively expensive (see Appendix A.1). We note that, for COLS, we can guarantee that the EMC of the best set will monotonically decrease over time, which we formally state below:

**Theorem 4.1** (Monotonicity of Cost-Optimized Local Search Algorithm)**.** Given the best set, $\mathcal{S}_{t-1}^{best} \in \mathbb{R}^{N \times d}$, the candidate set at iteration $t$, $\mathcal{S}_t \in \mathbb{R}^{N \times d}$, the matrix $\mathbf{C}^b \in \mathbb{R}^{N \times M}$ and $\mathbf{C} \in \mathbb{R}^{N \times M}$ containing the incurred cost of each counterfactual in $\mathcal{S}_{t-1}^{best}$ and $\mathcal{S}_t$ with respect to all the $M$ sampled cost functions $\{\mathcal{C}_i\}_{i=1}^M$, there always exist a $\mathcal{S}_t^{best}$ constructed from $\mathcal{S}_{t-1}^{best}$ and $\mathcal{S}_t$ such that

$$\text{ExpMinCost}(\boldsymbol{s}_u, \mathcal{S}_t^{best}; \{\mathcal{C}_i\}_{i=1}^M) \leq \text{ExpMinCost}(\boldsymbol{s}_u, \mathcal{S}_{t-1}^{best}; \{\mathcal{C}_i\}_{i=1}^M)$$

For a formal proof of the theorem, please refer to Appendix A.2.2.

**P-COLS:** The method P-COLS is a variant of COLS which starts multiple parallel runs of COLS with different initial sets. The run with the least objective value is selected as recourse for the user.

**Random Search:** We use Random Search as a baseline, which selects random counterfactuals from the whole space to obtain the next candidate set at each iteration.

| Data | Method | Metrics | | | | | | |
|------|--------|---------|---|---|---|---|---|---|
| | | Cost Metrics | | | Distance Metrics | | | |
| | | **FS@1** | **PAC($\downarrow$)** | **Cov** | **Div** | **Prox** | **Spars** | **Val** |
| | DICE | 2.47 | 1.37 | 8.32 | 3.90 | 65.80 | 47.20 | 97.90 |
| | Face-Eps | 15.23 | 0.76 | 22.60 | 4.75 | **92.22** | 74.98 | **100** |
| | Face-Knn | 25.30 | 0.74 | 35.00 | 8.62 | 89.07 | 71.85 | **100** |
| Adult-Income | Act. Recourse | 49.93 | 0.55 | 56.85 | 18.38 | 74.68 | 73.57 | 78.67 |
| | Random | 6.27 | 1.40 | 31.83 | **48.30** | 55.83 | 39.85 | 95.55 |
| | COLS | 72.57 | **0.38** | 76.07 | 25.77 | 80.22 | 76.48 | 97.15 |
| | P-COLS | **75.82** | 0.40 | **79.20** | 25.57 | 81.67 | **78.00** | 94.78 |
| | DICE | 0.40 | 0.54 | 0.40 | 11.30 | 65.00 | 32.00 | 98.90 |
| | Face-Eps | 12.20 | 0.29 | 12.20 | 2.50 | **94.20** | 60.60 | **100** |
| | Face-Knn | 12.20 | 0.29 | 12.20 | 2.60 | 94.10 | 60.60 | **100** |
| COMPAS | Act. Recourse | 65.80 | 0.40 | 66.60 | 11.87 | 80.53 | **74.07** | 44.23 |
| | Random | 29.95 | 0.77 | 39.20 | **42.22** | 55.90 | 31.25 | 71.88 |
| | COLS | 82.23 | **0.24** | 82.23 | 29.32 | 77.82 | 70.05 | 95.48 |
| | P-COLS | **83.73** | **0.24** | **83.73** | 29.38 | 78.48 | 71.30 | 92.78 |

Table 1: Table comparing different recourse methods across various cost and distance metrics (Section 5.1). The numbers reported are averaged across 5 different runs. For all the metrics higher is better except for PAC where lower is better. Refer to section 5.2 for more details.

# 5 EXPERIMENTS

## 5.1 EXPERIMENTAL SETUP

**Dataset:** We conduct our experiments on the Adult-Income (Dua & Graff, 2017) and COMPAS (Larson et al., 2016) datasets. The Adult-Income dataset is based on the 1994 Census database, and it contains 12 features. The model's task is to classify whether the income of an individual is over $50,000$. COMPAS was collected by ProPublica, and it contains information about the criminal history of defendants from Broward County for analyzing recidivism in the United States. The processed dataset contains 7 features. A model needs to decide bail based on predicting which of the bail applicants will recidivate in the next two years. We preprocess both datasets based on a previous analysis where all categorical features are mapped onto two classes (Pawelczyk et al., 2021). [1] Our black-box model is a neural network model with 2-layers. Please refer to Appendix B.1 and Table 4 for more details about experiments, data statistics, and the black-box model.

**Baselines:** We compare our methods COLS and P-COLS with Random Search, DICE (Mothilal et al., 2020), FACE-Knn and FACE-Epsilon (Poyiadzi et al., 2020), and Actionable Recourse (Ustun et al., 2019). Importantly, **we control for compute** by restricting the number of forward passes to the black-box model, which are needed to decide if a counterfactual produces the desired class. Additionally, for most big models this is the rate limiting step. For a description of the objective function and other details of these methods refer to Appendix A.1.1, A.2.3.

**Metrics:** To compare with past work, we evaluate methods on Distance based metrics like diversity, proximity, sparsity and validity metrics. Proximity is defined as $prox(\boldsymbol{x}, \mathbf{Y}) = 1 - \frac{1}{N} \sum_{i=1}^{|\mathbf{Y}|} dist(\boldsymbol{x}, \mathbf{Y}_i)$, where $\mathbf{Y}_i$ is a counterfactual. The Sparsity metric (Mothilal et al., 2020) is defined as $spar(\boldsymbol{x}, \mathbf{Y}) = 1 - \frac{1}{Nd} \sum_{i=1}^{|\mathbf{Y}|} \sum_{j=1}^{|\boldsymbol{x}|} \mathbb{1}_{\{x_j \neq \mathbf{Y}_{ij}\}}$. The Diversity metric (Mothilal et al., 2020) is defined as $div(\mathbf{Y}) = -\frac{1}{Z} \sum_{i=1}^{|\mathbf{Y}|-1} \sum_{j=i+1}^{|\mathbf{Y}|} dist(\mathbf{Y}_i, \mathbf{Y}_j)$, where $Z$ is the number of terms in the summation. Validity is defined as $val(Y) = \frac{|\{\text{unique } s_i \in \mathcal{S} : f(s_i) = +1\}|}{|\mathcal{S}|}$. Additionally, we show results on cost based metrics like Fraction Satisfied (FS@k), Population Coverage (Cov) (see Section 3) and the average incurred cost by the users (PAC) covered by the method.

**Simulated Cost Functions:** Given true user cost functions $\mathcal{C}_u^* \sim \mathcal{D}^*$ are not known in practice, hence for the experiments we simulated them from a distribution $\mathcal{D}_{test}$ for method evaluation. There are two phases in the experiments, the recourse generation phase where we optimize for EMC and the cost functions required are sampled $\mathcal{D}_{train}$. In the evaluation phase, we use the simulated user cost function which is hidden during training to compute cost-based metrics for all the re-

---

[1] The code for the Actionable Recourse method (Ustun et al., 2019) requires binary categorical variables.

| Optim. Method | Objective | Cost Metrics | | | Distance Metrics | | |
|---|---|---|---|---|---|---|---|
| | | FS@1 | PAC($\downarrow$) | Cov | Diversity | Proximity | Sparsity |
| LS | Sparsity | 10.1 | 1.304 | 29.0 | 42.7 | 66.2 | 55.8 |
| LS | Proximity | 9.7 | 1.275 | 27.0 | 42.1 | 67.5 | 55.0 |
| LS | Diversity | 0.0 | 2.393 | 7.6 | **53.3** | 50.8 | 35.6 |
| LS | EMC | 49.8 | 0.597 | 59.1 | 37.8 | 73.3 | 67.5 |
| COLS | EMC | **68.8** | **0.391** | **72.6** | 27.1 | **77.5** | **73.5** |

Table 2: Ablation results with Search algorithms trained on different objectives (Section 5.2).

course methods (see Appendix B.1.2). For all experiments apart from distribution shift experiments, $\mathcal{D}_{train} = \mathcal{D}_{test} = \mathcal{D}_{mix}$.

## 5.2 RESEARCH QUESTIONS

**Q1. Which Method Satisfies the Most Users?**

**Design:** In this experiment, we compare different recourse methods (Section 5.1 and Appendix A.2.3) with respect to distance and cost-based metrics (Section 5.1). We set a fixed budget of 5000 and a set size $|\mathcal{S}| = 10$ for all the methods. We perform five runs for each method with different random seeds and report the mean results in Table 1. We omit the variances from the table as 97% of the values have variance less than 0.01%, with the maximum value being 3.3% for validity of Random Search on COMPAS dataset.

**Results:** We can see from these results that DICE, which optimizes for a combination of distance-based metrics, performs much worse on the direct metrics like Population Coverage (Cov) and FS@k. On the other hand, **COLS and P-COLS, which optimize for EMC, achieve 22.64% and 25.89% point higher user satisfaction while covering 19.28% and 22.42% point more users** on the Adult-Income dataset, with similar improvements on COMPAS. We also observe that **COLS and P-COLS demonstrate high sparsity and proximity in the solutions**. Regarding recourse diversity, we find that **COLS and P-COLS do not exhibit high diversity**. The second-best method on cost metrics, Actionable Recourse, also promotes proximate and sparse solutions rather than diverse options, while Random Search achieves high diversity but low FS@1. These results provide evidence that diversity is not a necessary condition for high user satisfaction. Hence, it is preferable to provide users with recourse based on expected cost as opposed to providing them diverse options which might not align well with their preferences. Lastly, we note that FACE does the best on proximity as it finds the shortest high-density path in the generated graph.

**Q2. Is the Performance Improved by the Optimization Method or by the Objective?**

**Design:** We perform an ablation study to attribute the improvements from our method to either the COLS optimization method or the EMC objective function. To do so, we run a basic local search (LS) to optimize for diversity, proximity, and sparsity along with validity. We use a basic non-optimized local search, because there is no simple and efficient way to guarantee a reduction in the diversity objective by swapping out single elements from the solution set (see Appendix A.1). For a fair comparison across these objectives, we also optimize for EMC using basic local search.

**Results:** We provide the results for this experiment in Table 2. These results suggest that optimizing for metrics besides EMC is sub-optimal. For diversity, proximity, and sparsity objectives, the FS@1 score and population coverage is very low, while they perform well on the respective metrics. The low FS@1 score for distance metrics is expected as they ignore user-specific preference while optimizing for their objectives, and therefore the generated recourses might be infeasible under the user's cost function. We find that EMC with LS outperforms all Distance Metrics with LS on FS@1, which suggests that the **EMC objective function leads to higher user satisfaction**. Meanwhile, the **19% point difference in the performance of EMC with LS and COLS can be attributed to our Cost Optimization** described in Section 4.3 and Theorem 4.1, which allows COLS to more efficiently search the solution space. In this experiment, we observe that COLS again does poorly on diversity, whereas it promotes proximity and sparsity in the recourse. Based on results from Q1 and Q2, we conclude that having **high diversity is neither a necessary or sufficient condition to satisfy individual users with inherent feature preferences**. Additionally, we observe that **sparsity and proximity are positively correlated with higher user satisfaction** and inherently emerge from the idea of satisfying more users.

| Method | Gender | FS@1 | Cov | DIR-FS | DIR-Cov | Race | FS@1 | Cov | DIR-FS | DIR-Cov |
|---|---|---|---|---|---|---|---|---|---|---|
| **DICE** | **F** | 0.0 | 0.0 | - | - | **NW** | 0.0 | 0.0 | - | - |
| | **M** | 4.7 | 15.6 | | | **W** | 3.1 | 10.4 | | |
| **Face-Eps** | **F** | 12.5 | 22.1 | 1.504 | 1.118 | **NW** | 7.7 | 12.7 | 2.312 | 2.047 |
| | **M** | 18.8 | 24.7 | | | **W** | 17.8 | 26.0 | | |
| **Face-Knn** | **F** | 29.9 | 36.3 | 0.719 | 0.89 | **NW** | 12.7 | 25.4 | 2.228 | 1.425 |
| | **M** | 21.5 | 32.3 | | | **W** | 28.3 | 36.2 | | |
| **Act. Recourse** | **F** | 53.8 | 58.7 | 0.881 | 0.959 | **NW** | 46.5 | 54.9 | 1.101 | **1.056** |
| | **M** | 47.4 | 56.3 | | | **W** | 51.2 | 58.0 | | |
| **Random** | **F** | 7.8 | 34.6 | 0.859 | 0.792 | **NW** | 4.9 | 28.9 | 1.571 | 1.076 |
| | **M** | 6.7 | 27.4 | | | **W** | 7.7 | 31.1 | | |
| **COLS** | **F** | 72.7 | 76.2 | 0.994 | 0.992 | **NW** | 67.6 | 71.1 | 1.089 | 1.082 |
| | **M** | 72.3 | 75.6 | | | **W** | 73.6 | 76.9 | | |
| **P-COLS** | **F** | 76.5 | 80.2 | **1.004** | **1.0** | **NW** | 72.5 | 74.6 | **1.07** | 1.092 |
| | **M** | 76.8 | 80.2 | | | **W** | 77.6 | 81.5 | | |

Table 3: Fairness analysis of recourse methods for subgroups with respect to Gender and Race. **DIR**: Disparate Impact Ratio; **M**: Male, **F**: Female; **W**: White, **NW**: Non-White (Section 5.2).

### Q3. Are Recourses Fair Across Subgroups?

**Design:** Here, we want to understand whether recourse methods provide equitable solutions across subgroups based on demographic features like *Gender* and *Race*. In this experiment, we assume Gender is non-actionable for the sake of analysis. Existing works on algorithmic fairness present a number of metrics for characterizing the disparate impact of black-box classification models across population subgroups (Feldman et al., 2015). We adapt the Disparate Impact metric for the recourse outcomes we study, which we denote by Disparate (DIR). Given a metric $\mathcal{M}$, DIR is a ratio between metric scores across two subgroups. DIR-$\mathcal{M} = \mathcal{M}(\text{S=1})/\mathcal{M}(\text{S=0})$. We use either Cov or FS@1 as $\mathcal{M}$. Under the DIR metric, a maximally fair method achieves a score of 1. We run experiments on the Adult-Income dataset, with a budget of 5000 and $|\mathcal{S}| = 10$.

**Results:** We present the subgroup-based results in Table 3. We observe that COLS and P-COLS typically have a lower difference in FS@1 and Cov across subgroups (*Gender* and *Race*) as compared to other baselines while satisfying and covering significantly more users. In particular, we find that our method achieves a score very close to 1 on DIR-FS and DIR-Cov for the gender-based subgroup, and scores relatively close to 1 on the race-based subgroup. We attribute the fairness of our method to (1) the fact that our recourses are individualized, rather than making use of the data distribution, and (2) the use of a diverse set of cost functions when generating recourses. We see condition (2) as important since there are other individualized methods that do not rely on the data distribution, such as Actionable Recourse, which can generate less fair solutions than COLS. Overall, we conclude that **our method is typically more fair than baselines on both Gender and Race-based subgroups while providing recourse to a larger fraction of people in both subgroups**.

### 5.3 ADDITIONAL RESULTS

We provide experiments for several additional research questions in the Appendix B.2, which we summarize here: (1) We find that our method is **robust to misspecification of the cost function distribution** (Figure 2, 3); (2) We can **make use of a larger compute budget to scale up the performance** (Figure 4); The **recourse sets provide high quality solutions to users using as few as 3 counterfactuals** (Figure 5); and (4) we can **achieve high user satisfaction with as few as 20 Monte Carlo samples, rather than 1000** (Figure 6). We also show some qualitative examples of recourses provided by our method in Table 5.

## 6 CONCLUSION

In this paper, we propose a cost-based recourse generation method which can optimize for an unknown user-specific cost function. With simulated cost functions, we show that our method achieves much higher rates of user satisfaction than comparable baselines. This is particularly promising since detailed user cost function data is not readily available for most applications. We attribute the method's efficacy to our Expected Minimum Cost (EMC) objective term, and we also show that our discrete optimization algorithm, Cost-Optimized Local Search (COLS), produces large improvements over baseline search methods. Lastly, we observe that our recourses are often more fair than baseline recourse methods, while offering recourse to a much larger fraction of the user population.

## 7 ETHICS STATEMENT

We hope that our recourse method is adopted by institutions seeking to provide reasonable paths to users for achieving more favorable outcomes under the decisions of black-box machine learning models or other inscrutable models. We see this as a "robust good," similar to past commentators Venkatasubramanian & Alfano (2020). Below, we comment on a few other ethical aspects of the algorithmic recourse problem.

First, we suggest that fairness is an important value which recourse methods should always be evaluated along, but we note that evaluations will depend heavily on the model, training algorithm, and training data. For instance, a sufficiently biased model might not even allow for suitable recourses for certain subgroups. As a result, any recourse method will fail to identify an equitable set of solutions for the population. That said, recourse methods can still be designed to be more or less fair. This much is evident from our varying results on fairness metrics using a number of recourse methods. What will be valuable in future work is to design experiments that separate the effects on fairness of the model, training algorithm, training data, and recourse algorithm. Until then, we risk blaming the recourse algorithm for the bias of a model, or vice versa.

Additionally, there are possible dual-use risks from developing stronger recourse methods. For instance, malicious actors may use recourse methods when developing models in order to *exclude* certain groups from having available recourse, which is essentially a reversal of the objective of training models for which recourse is guaranteed (Ross et al., 2021). We view this use case as generally unlikely, but pernicious outcomes are possible. We also note that these kinds of outcomes may be difficult to detect, and actors may make bad-faith arguments about the fairness of their deployed models based on other notions of fairness (like whether or not a model has access to protected demographic features) that distract from an underlying problem in the fairness of recourses.

## 8 REPRODUCIBILITY STATEMENT

To encourage reproducibility, we provide our source code, including all the data pre-processing, model training, recourse generation, and evaluation metric scripts in the supplementary material. The details about the datasets and the pre-processing is provided in Appendix B.1.1. We also provide clear and concise Algorithms 1, 4, 5 for our cost sampling procedures and our optimization method COLS in Algorithm 2. Additionally, we also provide formal proof of the Theorem 4.1 stated in paper in Appendix A.2.2 along with the constructive procedure for the proof is provided in Algorithm 2.

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

## A  APPENDIX - OBJECTIVE AND OPTIMIZATION

### A.1  PROPOSED METHOD

#### A.1.1  OTHER OBJECTIVES

To obtain feasible a counterfactual set, past works have used various objective terms. We list objectives below from methods we compare with.

**1.  DICE** (Mothilal et al., 2020) optimizes for a combination of Distance Metrics like *diversity* and *proximity*. They model diversity via Determinantal Point Processes (Kulesza & Taskar, 2012) adopted for solving subset selection problems with diversity constraints. They use determinant of the kernel matrix given by the counterfactuals as their diversity objective as defined below.

$$dpp\_diversity(\mathcal{S}) = det(\mathbf{K}), \text{ where} \mathbf{K}_{ij} = \frac{1}{1 + dist(\boldsymbol{s}_i, \boldsymbol{s}_j)}$$

Here, $dist(\boldsymbol{s}_i, \boldsymbol{s}_j)$ is the normalized distance metric as defined in Wachter et al. (2017) between two state vectors. *Proximity* is defined in terms of the distance between the original state vector and the counterfacutals, $prox(\boldsymbol{x}, \mathbf{Y}) = 1 - \frac{1}{N}\sum_{i=1}^{|\mathbf{Y}|} dist(\boldsymbol{x}, \mathbf{Y}_i)$, where $\mathbf{Y}_i$ is a counterfactual.

**2. Actionable Recourse** (Ustun et al., 2019) work under the assumption that all features have equal preference scores for all the users. They define cost function based on the log-percentile shift is given by,

$$cost(\boldsymbol{s} + a; \boldsymbol{s}) = \sum_{j \in \mathcal{J}_A} \log \frac{1 - Q_j(\boldsymbol{s}_j + a_j)}{1 - Q_j(\boldsymbol{s}_j)}$$

where $Q_j(.)$ is the cumulative distribution function of $\boldsymbol{s}_j$ in the target population, $\mathcal{J}_A$ is the set of actionable features and $a_j$ is the action performed on the feature $j$.

---

**Algorithm 1:** Procedure for sampling Cost Functions.

---

**Input:** A state vector $s$, **Optional:** Preferred featured $\mathcal{F}_p$, feature preference scores $p$, cost distribution mixing weight $\alpha$

**Output:** Sample preference scores $p$ and the cost functions $\mathcal{C}$.

1 **function** sampleCostAndPreference($s, \alpha = None \mathcal{F}_p = \{\}, p = None$)

2     **if** $\mathcal{F}_p$ *is empty* **then**

3         | $\mathcal{F}_p \sim RandomSubset(\mathcal{F}_m)$           ▷ Sample preferred features.

4

5     **if** $p$ *is None* **then**

6         | $concentration = [1 \; if \; f \in \mathcal{F}_p \; else \; 0 \; for \; f \; in \; \mathcal{F}]$

7         | $p \sim Dirichlet(concentration)$     ▷ Sample feature preference scores.

8

9     **if** $\alpha$ *is None* **then**

10        | $\alpha \sim Uniform(0,1)$            ▷ Sample cost mixing weight.

11

12     $\mathcal{C} = \{\}$

13     **forall** $f_i \in \mathcal{F}$ **do**

14         $\mu^{(f_i, Lin)}, \sigma^{(f_i, Lin)} \longleftarrow getLinearCostMean(s, p^{(f_i)}, f_i, \mathcal{F}_p)$

15         $\mu^{(f_i, Perc)}, \sigma^{(f_i, Perc)} \longleftarrow getPercentileCostMean(s, p^{(f_i)}, f_i, \mathcal{F}_p)$

16         $\mu^{(f_i)} \longleftarrow \alpha * \mu^{(f_i, Lin)} + (1 - \alpha) * \mu^{(f_i, Perc)}$

17         $\sigma^{(f_i)} \longleftarrow \alpha * \sigma^{(f_i, Lin)} + (1 - \alpha) * \sigma^{(f_i, Perc)}$

18         $\mathcal{C}^{(f_i)} \longleftarrow Beta(\mu^{(f_i)}, \sigma^{(f_i)})$     ▷ Beta with mean and variance $\mu^{(f_i)}$, $\sigma^{(f_i)}$

19         $\mathcal{C} \leftarrow \mathcal{C} \cup \mathcal{C}^{(f_i)}$

20     **return** $p, \mathcal{C}_p$

---

**Algorithm 2:** Cost-Optimized Local Search Algorithm

---

**Input:** A state vector $s$, $\{\mathcal{C}_i\}_{i=1}^M \sim \mathcal{D}_u$ cost distributions

**Output:** $\mathcal{S}^{best}$, a set of generated counterfactuals of size $N$.

1 **function** LocalSearch($s, \{\mathcal{C}_i\}_{i=1}^M, hammingDistance = 2$)

2     **Initialize**

3         $\mathcal{S}^{best} \in \mathbb{R}^{N \times d} \leftarrow$ pertubCFS($s, hammingDistance$)     ▷ Perturb $s$, $N$ times.

4         $\mathbf{C}^b \leftarrow$ getCostMatrix($s, \mathcal{S}^{best}; \{\mathcal{C}_i\}_{i=1}^M$)     ▷ Incurred costs for $\mathcal{S}^{best}$.

5     **while** *usedBudget < Budget* **do**

6         $\mathcal{S} \in \mathbb{R}^{N \times d} \leftarrow$ pertubCFS($\mathcal{S}^{best}, hammingDistance$)

7         $\mathbf{C} \in \mathbb{R}^{N \times M} \leftarrow$ getCostMatrix($s, \mathcal{S}; \{\mathcal{C}_i\}_{i=1}^M$)     ▷ Incurred costs for the $\mathcal{S}$.

        // $\mathbf{B}_{ij}$ = Change in objective when $\mathcal{S}^{best}[i] \leftarrow \mathcal{S}[j]$.

8         $\mathbf{B} \in \mathbb{R}^{N \times N} \leftarrow$ computeBenefits($\mathbf{C}^b, \mathbf{C}$)     ▷ Refer to Algorithm 3

        // Greedily select which pairs to swap given $\mathbf{B}$

9         replaceIndices $\leftarrow$ getReplaceIdx($\mathbf{B}$)

        // Swap these pairs and update $\mathbf{C}^b$.

10         **forall** *originalIdx, replaceIdx* $\in$ *replaceIndices* **do**

11             | $\mathcal{S}^{best}[originalIdx] = \mathcal{S}[replaceIdx]$

12         $\mathbf{C}^b \leftarrow$ getCostMatrix($s, \mathcal{S}^{best}; \{\mathcal{C}_i\}_{i=1}^M$)

13     **return** $\mathcal{S}^{best}, \mathbf{C}^b$

---

---

**Algorithm 3:** Algorithm for Theorem 4.1

---

**Input:** $\mathbf{C}^b, \mathbf{C} \in \mathbb{R}^{N \times M}$ matrices containing the costs with respect to all cost samples..
**Output:** $\mathbf{B} \in \mathbb{R}^{N \times N}$, matrix containing the benefits of replacing pairs from $\mathcal{S}_{t-1}^{best} \times \mathcal{S}_t$

1 **function** computeBenefits($\mathbf{C}^b, \mathbf{C}$)

2    **Initialize**

3        $\mathbf{B} \in \mathbb{R}^{N \times N} \leftarrow \mathbf{0}$

      `// Find the indices of the best and second best counterfactual in`
          `𝒮ᵇᵉˢᵗ for each of the M cost function.`

4    $\boldsymbol{b}^1 \in \mathbb{R}^M = \arg\max_i \mathbf{C}_{ij}^b$

5    $\boldsymbol{b}^2 \in \mathbb{R}^M = \arg \text{ second } \max_i \mathbf{C}_{ij}^b$

      `// Iterate over all pairs of counterfactuals.`

6    **forall** $p, q \in [N] \times [N]$ **do**

         `// Iterate over cost functions for which` $p^{th}$ `counterfactual in`
            `𝒮ᵇᵉˢᵗ has the minimum cost.`

7       **forall** $r \in \{i \in [M] \mid \boldsymbol{b}_i^1 = p\}$ **do**

8          **if** $\mathbf{C}_{pr}^b > \mathbf{C}_{qr}$ **then**

            `// This replacement reduces the cost of` $\mathcal{S}^{best}$ `by` $\mathbf{C}_{pr}^b - \mathbf{C}_{qr}$.

9             $\mathbf{B}_{pq} += \mathbf{C}_{pr}^b - \mathbf{C}_{qr}$

10         **else**

            `//` $\mathbf{C}_{\boldsymbol{b}_r^2,r}^b$ `= cost of second best counterfactual in` $\mathcal{S}^{best}$ `for` $r^{th}$
              `cost function.`

11             $\mathbf{B}_{pq} += \mathbf{C}_{pr}^b - min(\mathbf{C}_{qr}, \mathbf{C}_{\boldsymbol{b}_r^2,r}^b)$

12    **return** $B$

---

### A.2   OPTIMIZATION METHODS

#### A.2.1   HIERARCHICAL COST SAMPLING PROCEDURE

To optimize for EMC, we need a plausible distribution which can model users' cost functions. We propose a hierarchical cost sampling distribution which provides cost samples that are a linear combination of *percentile shift cost* (Ustun et al., 2019) and *linear cost*, where the weights of this combination are user-specific. *Percentile shift cost* for ordered features is proportional to the change in a feature's percentile associated with the change from an old feature value to a new one. E.g., if a user is asked to increase the number of work hours from 40 to 70, then given the whole dataset, we can estimate the percentile of users working 40 and 70 hours a week. The cost incurred is then proportional to the difference in these percentiles. The *Linear cost* for ordered features is proportional to the number of intermediate states in $Q^{(f)}$ which a user will have to go through while transitioning from their current state to the final state. E.g., if a user is asked to change their education level from *High-school* to *Masters* then there are two steps involved in the process. First, they need to get a *Bachelors* degree and then a *Masters* degree in which case, the user's cost is proportional to 2 because of the two steps involved in the process. To sample a cost function $\mathcal{C} = \{\mathcal{C}^{(f)} \in \mathbb{R}^{|Q^{(f)}| \times |Q^{(f)}|} \mid \forall f \in \mathcal{F}\}$, we independently sample $\mathcal{C}^{(f)}$ for each feature (see Algorithm 1). We first randomly sample a subset of editable features for the user, and then we sample feature preference score $\boldsymbol{p}_u$ from a Dirichlet distribution with a uniform prior over selected features. These will be used to scale costs, such that a higher value of $\boldsymbol{p}_u^{(f)}$ implies a lower transition cost for feature $f$. For both the percentile and linear cost, the cost $\mathcal{C}_{ij}^{(f)}$ of transitioning from feature state $i \rightarrow j$, is sampled from a Beta distribution on the interval $[0, 1]$. The mean of this distribution depends on the types of cost (linear or percentile) and the feature type (ordered or unordered). Here, we first obtain one mean for linear cost ($\mu_{ij}^{(f,lin)}$) and one mean for percentile cost ($\mu_{ij}^{(f,perc)}$) and then combine them to form a single Beta mean. Each of the two means is proportional to the change in the feature (in either linear or percentile terms) when the feature is ordered. For unordered features, the mean is randomly sampled from

the unit interval (see Algorithms 5, 4). Then, the linear and percentile means are multiplied with $(1 - \boldsymbol{p}_u^{(f)})$ to scale the transition cost according to the feature preference score. Next, the two means are combined to obtain a single mean, $\mu_{ij}^{(f)} = \alpha * \mu_{ij}^{(f,lin)} + (1 - \alpha) * \mu_{ij}^{(f,perc)}$, where $\alpha \in [0, 1]$ represents whether a user thinks of cost in terms of linear or percentile shift (sampled randomly from unit interval). Note that the value $\mu_{ij}^{(f)}$ are monotonic, i.e. if the user has to make more drastic changes to the feature, then the associated cost will be higher. The variance for the Beta distribution, $\sigma_{ij}^{(f)}$, is set to constant value of 0.01. Finally, the cost $\mathcal{C}_{ij}^{(f)}$ is sampled from $\text{Beta}(\mu_{ij}^{(f)}, \sigma_{ij}^{(f)})$. We emphasize that this sampling procedure allows users to partially specify their cost functions, e.g. by denoting which features they prefer to edit (the Dirichlet mean) or the relative difficulty of editing one feature versus another ($\boldsymbol{p}_u$). If we set $\alpha = 0$, then the resulting distribution is $\mathcal{D}_{perc}$ and with $\alpha = 1$ we get the distribution $\mathcal{D}_{lin}$.

### A.2.2 MERGING COUNTERFACTUAL SETS

When searching for a good solution set, it would be useful to have the option of improving on the best set we have obtained so far using individual counterfactuals in the next candidate set we see, rather than waiting for a new, higher-scoring set to come along. While optimizing for objectives like diversity, which operate over all pairs of elements in the set, it is computationally complex to evaluate the change in the objective function if one element of the set is replaced by a new one. To evaluate the change in objective in such cases, we need iterate over all pairs of element in the best and the candidate set and then evaluate the objective for the whole set again. The iteration over both the sets here is not the hard part but the computation that needs to be done within. For our objective, we can compute costs for individual recourses rather than sets, meaning we can do a trivial operation to compute the benefits of each pair replacement. But, if we wanted to do this with diversity then for each pair of replacement we need to compute additional $\mathcal{S}$ distances for each replacement because the distance of the new replace vector needs to be computed with respect to all the other vectors, for each iteration of the nested loop. This quickly makes it infeasible to improve the best set by replacing individual candidates with the best set elements. However, for metrics where it is easy to evaluate the effect of individual elements on the objective function, we can easily merge the best set and any other set $\mathcal{S}_t$ from time $t$ to monotonically increase the objective function value.

In our objective function, EMC, we can compute the goodness of individual counterfactuals with respect to all the Monte Carlo samples (Robert & Casella, 2010). Given a set of counterfactuals we can obtain a matrix of incurred cost $\mathbf{C} \in \mathbb{R}^{N \times M}$, which specifies the cost of each counterfactual for each of the Monte Carlo samples. We can use this to update the best set $\mathcal{S}^{best}$ using elements from the perturbed set $\mathcal{S}_t$ at time $t$. This procedure is defined in algorithm 3. It iterates over all pairs of element in $\boldsymbol{s}_i \in \mathcal{S}^{best}$ and $\boldsymbol{s}_j \in \mathcal{S}_t$ and computes the change that will occur in the objective function by replacing $\boldsymbol{s}_i \to \boldsymbol{s}_j$. Note that we are not recomputing the costs here. Given $\mathcal{S}^{best}, \mathcal{S}_t$, $\mathbf{C}^b$ and $\mathbf{C}$, we can guarantee that we will update the best set $\mathcal{S}_{best}$ in a way to improve the mean of the minimum costs incurred for all the Monte Carlo samples. This is shown in algorithm 3 and the monotonicity of the EMC objective under this case can be formally stated as,

**Theorem** (Monotonicity of Cost-Optimized Local Search Algorithm). Given the best set, $\mathcal{S}_{t-1}^{best} \in \mathbb{R}^{N \times d}$, the candidate counterfactual at iteration $t$, $\mathcal{S}_t \in \mathbb{R}^{N \times d}$, the matrix $\mathbf{C}^b \in \mathbb{R}^{N \times M}$ and $\mathbf{C} \in \mathbb{R}^{N \times M}$ containing the incurred cost of each counterfactual in $\mathcal{S}_{t-1}^{best}$ and $\mathcal{S}_t$ with respect to all the $M$ sampled cost functions $\{\mathcal{C}_i\}_{i=1}^{M}$, there always exist a $\mathcal{S}_t^{best}$ constructed from $\mathcal{S}_{t-1}^{best}$ and $\mathcal{S}_t$ such that

$$\text{ExpMinCost}(\boldsymbol{s}_u, \mathcal{S}_t^{best}; \{\mathcal{C}_i\}_{i=1}^{M}) \leq \text{ExpMinCost}(\boldsymbol{s}_u, \mathcal{S}_{t-1}^{best}; \{\mathcal{C}_i\}_{i=1}^{M})$$

*Proof.* To prove this theorem, we construct a procedure that ensures that the ExpMinCost is monotonic. For this procedure, we prove that the monotonicity of EMC holds. Check algorithm 3 for a constructive procedure for this proof, which is more intuitive to understand.

We start off by noting that each element of $\mathbf{C}_{ij}^b$ is the cost of the $i^{th}$ counterfactual $\boldsymbol{s}_i^b$ in the best set $\mathcal{S}_{t-1}^{best}$ with respect to the cost function $\mathcal{C}_j$ given by $\text{Cost}(\boldsymbol{s}_u, \boldsymbol{s}_i^b; \mathcal{C}_j)$. Similarly $\mathbf{C}_{ij} = \text{Cost}(\boldsymbol{s}_u, \boldsymbol{s}_i; \mathcal{C}_j)$ where $\boldsymbol{s}_i$ is the $i^{th}$ candidate counterfactual. Note that, the EMC is the average of the MinCost with respect to all the sampled cost function $\mathcal{C}_j$. What this means is that given a pair of counterfactual from $\mathcal{S}_{t-1}^{best} \times \mathcal{S}_t$ and for each $\mathcal{C}_j$, we can compute the change in the MinCost which we describe

later. These replacements can lead to an increase in the cost with respect to certain cost function but the overall reduction depend on the aggregate change over all the cost functions. Given this, for each replacement candidate pair in $\mathcal{S}_{t-1}^{best} \times \mathcal{S}_t$, we can compute the change in EMC by summing up the changes in the MinCost across all cost functions $\mathcal{C}_j$ ; this is called the cost-benefit for this replacement pair. The cost benefit can be negative for certain replacements as well if the candidate counterfactual increases the cost across all the cost functions. The pairs with the highest positive cost benefits are replaced to construct the set $\mathcal{S}_t^{best}$, if no pair has a positive benefit then we keep set $\mathcal{S}_{t-1}^{best} = \mathcal{S}_t^{best}$. Hence, this procedure monotonically reduces EMC. We now specify how the change in MinCost can be computed to complete the proof.

To compute the change in MinCost for a single cost function $\mathcal{C}_i$, first we find the counterfactual in $\mathcal{S}_{t-1}^{best}$ with the lowest and second lowest cost which we denote by $s_{l_1}^b$ and $s_{l_2}^b$. These are the counterfactuals which can affect the MinCost with respect to a particular cost function $\mathcal{C}_i$. This is true because when we replace the counterfactual $s_{l_1}^b$ which has the lowest cost for $\mathcal{C}_j$ with a new candidate counterfactual $s_i$, there are two cases. Either, $\mathbf{C}_{l_1 j}^b > \mathbf{C}_{ij}$ or $\mathbf{C}_{l_1 j}^b \leq \mathbf{C}_{ij}$. In case when the candidate $s_i$ has lower cost for $\mathcal{C}_j$ than $\mathbf{C}_{l_1 j}^b$, i.e. $\mathbf{C}_{l_1 j}^b > \mathbf{C}_{ij}$, then the replacement reduces the cost by $\mathbf{C}_{l_1 j}^b - \mathbf{C}_{ij}$. In case when the candidate cost for $\mathcal{C}_j$, $\mathbf{C}_{ij}$, is higher than the lowest cost in the best set $\mathbf{C}_{l_1 j}^b$, i.e. $\mathbf{C}_{l_1 j}^b \leq \mathbf{C}_{ij}$, it means that this replacement will increase the cost for $\mathcal{C}_i$ by $\mathbf{C}_{l_1 j}^b - min(\mathbf{C}_{ij}, \mathbf{C}_{l_2 j}^b)$. Here, $\mathbf{C}_{l_2 j}^b$ is the second lowest cost counterfactual for $\mathcal{C}_i$. Note that the change in this case will be negative and also depend on the second best counterfactual because once the $s_{l_1}^b$ is removed from the set, the best cost for $\mathcal{C}_i$ will either be for $s_{l_2}^b$ or $s_i$, hence we take the minimum of those two and then take the difference as the increase in cost. Please refer to Algorithm 3 for a cognitively easier way to understand the proof. $\qquad \square$

### A.2.3 Other Methods

In this section, we describe some of the optimization methods used by relevant baselines.

**1. DICE** (Mothilal et al., 2020) perform gradient-based optimization in this continuous space while optimizing for objective defined in Section A.1.1. Their final objective function is defined as

$$C(\boldsymbol{x}) = \underset{\boldsymbol{c}_1,...,\boldsymbol{c}_k}{\arg\min} \frac{1}{k} \sum_{i=1}^{k} \text{loss}(f(\boldsymbol{c}_i), y) + \frac{\lambda_1}{k} \sum_{i=1}^{k} \text{dist}(\boldsymbol{c}_i, \boldsymbol{x}) - \lambda_2 \, \text{dpp\_diversity}(\boldsymbol{c}_1, \ldots, \boldsymbol{c}_k)$$

where $\boldsymbol{c}_i$ is a counterfactual, $k$ is the number of counterfactuals, f(.) is the black box ML model, yloss(.) is the metric which minimizes the distance between models prediction and the desired outcome $y$. dpp_diversity(.) is the diversity metric as defined in Section A.1.1 and $\lambda_1$ and $\lambda_2$ are hyperparameters to balance the components in the objective. Please refer to Mothilal et al. (2020) for more details.

**2. FACE** (Poyiadzi et al., 2020) operates under the idea that to obtain actionable counterfactuals they need to be connected to the user state via paths that are probable under the original data distribution aka high-density path. They construct two different types of graphs based on nearest neighbors (Face-knn) and the $\epsilon$-graph (Face-Eps). They define geodesic distance which trades-off between the path length and the density along this path. Lastly, they use the Shortest Path First Algorithm (Dijkstra's algorithm) to get the final counterfactuals. Please refer to (Poyiadzi et al., 2020) for more details.

**3. Actionable Recourse** (Ustun et al., 2019) tries to find an action set $\boldsymbol{a}$ for a user such that taking the action changes the black-box models decision to the desired outcome class, denoted by $+1$. They try to minimize the cost incurred by the user while restricting the set of actions within an action set $A(x)$. The set $A(x)$ imposes constraints related to feasibility and actionability with respect to features. They optimize the log-percentile shift objective (see Section A.1.1). Their final optimization equation is

$$\min cost(\boldsymbol{a}; \boldsymbol{x}) \ \ s.t. \ \ f(\boldsymbol{x} + \boldsymbol{a}) = +1, \, \boldsymbol{a} \in A(\boldsymbol{x})$$

|                        | Adult-Income Binary | COMPAS Binary    | Adult-Income    | COMPAS           |
| ---------------------- | ------------------- | ---------------- | --------------- | ---------------- |
| # Continuous features  | 3                   | 4                | 2               | 3                |
| # Categorical features | 9                   | 3                | 10              | 12               |
| Undesired class        | $\leq$ 50k          | Will Recidivate  | $\leq$ 50k      | Will Recidivate  |
| Desired class          | > 50k               | Won't Recidivate | > 50k           | Won't Recidivate |
| Train/val/test         | 20088/2338/749      | 1415/229/491     | 13172/1569/748  | 5491/705/444     |
| Model Type             | ANN(2, 20)          | ANN(2, 20)       | ANN(2, 20)      | ANN(2, 20)       |
| Val Accuracy           | 82%                 | 69%              | 81%             | 61%              |

Table 4: Table containing data statistics and black-box model details. The binary version of the datasets are take from (Pawelczyk et al., 2021) whereas the non-binary version are taken from (Mothilal et al., 2020).

which is cast as an Integer Linear Program (Mittleman, 2018) to provide users with recourses. Their publicly available implementation is limited to a binary case for categorical features,[2] hence we demonstrate results on the binarized version of the dataset.

# B    APPENDIX - EXPERIMENTS AND DETAILS

## B.1    EXPERIMENTAL SETUP

### B.1.1    DATASETS AND BLACK-BOX MODEL

In our experiments, we have two versions of the dataset, one with binary categorical features, whereas the other with non-binary categorical features. In the main paper, we show results on the binarized version (Table 1) as an important baseline, Actionable Recourse (Ustun et al., 2019), operates with binary categorical features.[3] The data statistics for all the datasets can be found in Table 4. In our experiments, for all the datasets, the features gender and race are considered to be immutable (Mothilal et al., 2020), since we perform subgroup analysis with these variables that would be rendered meaningless if users could switch subgroups. Other features can either be mutable or conditionally mutable depending on semantics. These constraints can be incorporated into the methods by providing a schema of feature mutability criterion. Our black-box model is a multi-layer perceptron model with 2 hidden layers trained on the trained set and validated on the dev set. The accuracy numbers are shown in Table 4. The test set which is used in the counterfactual generation experiments only contains users which are classified to the undesired class by the trained black-box model. Note that our frameworks can operate with any type of model, the only requirement is the ability to query the model for outcome given a user's state vector.

### B.1.2    RECOURSE GENERATION AND EVALUATION PIPELINE

To approximate the expectation in equation 5, our algorithm samples a set of random cost functions $\{\mathcal{C}_i\}_{i=1}^{M} \sim \mathcal{D}_{train}$, which are used at the generation time to optimize for the user's hidden cost function. In the generation phase, we use Equation 6 as our objective. Note that, this objective promotes that the generated counterfactual set contains at least one good counterfactual for each of the cost samples, hence this set satisfies a large variety of samples from $\mathcal{D}_{train}$. This is achieved via minimizing the mean of the minimum cost incurred for each of the Monte Carlo samples (Robert & Casella, 2010). Equivalently, the objective is minimized by a set of counterfactuals $\mathcal{S}$ where for each cost function there exists an element in $\mathcal{S}$ which incurs the least possible cost. In practice the size of set $\mathcal{S}$ is restricted, hence we may not achieve the absolute minimum cost but the objective tries to ensure that the counterfactuals which belong to the set have a low cost at least with respect to one Monte Carlo cost sample. The generation phase outputs a set of counterfactuals $\mathcal{S}$ which is to be provided to the users as recourse options. Given this set $\mathcal{S}_u$, in the evaluation phase, we use the users simulated cost functions which are hidden in the generation phase, to compute the cost incurred by the user as defined in Equation 1 and calculate the metrics defined in the Section 5.1.

---

[2]Please refer to the this example where they mention about these restricted abilities https://github.com/ustunb/actionable-recourse/blob/master/examples/ex_01_quickstart.ipynb

[3]The binary datasets can be downloaded from https://github.com/carla-recourse/cf-data, whereas the non-binary data can be found on https://github.com/interpretml/DiCE.

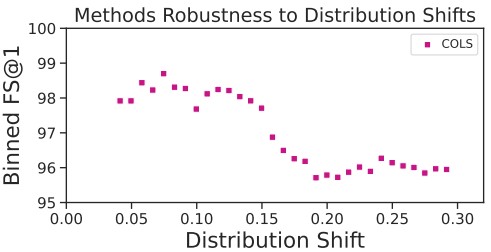

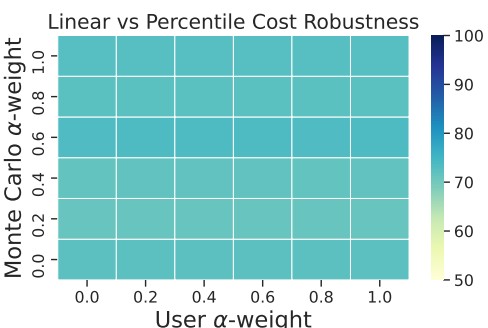

Figure 2: In this plot we show the fraction of users satisfied vs the distance between the train and test distributions. The results demonstrate that as the distance increases the performance drops a bit and then plateaus, which means that the method is robust to this kind of distribution shift. Please refer to Section B.2 for more details.

Figure 3: This figure shows the performance of the method on FS@k when recourses are generated with Monte Carlo cost samples from a distribution with $\alpha$-weight varying between 0 and 1, where the user costs follow different $\alpha$-weight values varying between 0, 1. Performance is robust to misspecification of $\alpha$. Refer to Section B.2 for more details.

**Remark 1.** In the limiting case, as $|\mathcal{S}| \to M$ and Budget $\to \infty$, RHS of equation 6; $\min_{\mathcal{S}} \mathrm{MinCost}(s_u, \mathcal{S}; \mathcal{C}_i) \to$ optimum cost, $\forall\, i \in [M]$, if feasible solutions are possible for all cost functions samples $\{\mathcal{C}_i\}_{i=1}^{M}$.

What this remark implies is that, as the set size increases to the number of Monte Carlo samples $M$, then the methods can trivially generate one counterfactual for each cost function sample which minimizes its cost. In that limiting case, we get custom counterfactual for each cost function. Hence, in the limiting case we can find optimum solution. In the experiment B.2, we show that even with very less number of counterfactual we are able to achieve good performance implying that in practice we do not need large set size.

### B.2 ADDITIONAL RESEARCH QUESTIONS

**Q4. Are Solutions Robust to Misspecified Cost Distributions?**

**Design:** In our hierarchical cost sampling procedure, we make minimal assumptions about the user's feature preferences if they are not provided by the user. When finding recourses, we select a random subset of features along with their preference score for each user. However, there are situations where user preferences may be relatively homogeneous for certain features where people usually share common preferences. For example, to increase their *income*, many users might prefer to edit their *occupation type* or *education level* rather than their *work hours* or *marital status*. Given the possibility of this kind of distribution shift in feature preferences, we want to measure how robust our method is to distribution shift between our hierarchical sampling distribution and the actual cost distribution followed by users.

In this experiment, we test a case of this kind of distribution shift over cost functions. For users in the Adult-Income data, we generate recourse sets using Monte Carlo samples from our standard distribution $\mathcal{D}_c$ (Algorithm 1). To obtain hidden user cost functions that differ from this distribution, we first generate 500 different feature subsets indicating which features are editable, where each subset corresponds to a binary vector *concentration* representing a user having specific preferences for some features over others (see Sec. 4.2 and Alg. 1). Since having different editable features induces a different distribution over cost functions, we obtain a measure of distribution shift for each of the 500 *concentration* vectors by taking an $l_2$ distance between the vector and *its nearest neighbor in the space of* concentration *vectors used to generate the recourses*. We use the nearest neighbor because the most outlying *concentration* vectors are least likely to be satisfied by the recourse set. In other words, the likelihood that a user is satisfied depends on the minimum distance between their *concentration* vector and its nearest neighbor in the cost samples used at recourse

generation time. Therefore, when the minimum distance increases, there is a greater distribution shift between the user's cost functions and those obtained from $\mathcal{D}_c$. Finally, we measure how many users are satisfied for a given degree of distribution shift.

**Results:** In Figure 2, we show a binned plot of FS@1 against our measure of distribution shift. We observe that as the distance between the distributions increases, the fraction of users satisfied decreases slightly and then plateaus. Even at the maximum distance we obtain, performance has only dropped about 3 points. This implies that **our method is robust to distribution shift in the cost distribution in terms of which features people prefer to edit.** We attribute this result to the fact that our method assumes random feature preferences and provides multiple recourse options, each of which can cater to different kinds of preferences. As a result, we achieve a good covering of the cost function space (see experiments w.r.t. varying recourse set size and number of sampled cost functions in the Appendix B.2).

**Q5. Robustness to Misspecification in Linear vs Percentile based costs?**

**Design:** Our sampling procedure samples cost by taking an $\alpha$-weighted combination of two different types of costs, linear and percentile costs. These two cost have different underlying assumptions about the how users view the cost of transition between the states. We want to test the robustness of our method in terms of misspecification in users disposition to these types of cost. We perform a robustness analysis where the users cost function has a different $\alpha$ mixing weight as compared to the Monte Carlo samples we use to optimize for EMC. This creates a distribution shift in the user cost function distribution and the Monte Carlo sampling distribution. We vary the user and Monte Carlo distributions $\alpha$-weights within the range of 0 to 1 in steps of 0.2. At the extremes values of $\alpha = 0, 1$, the shifts are very drastic as the underlying distribution changes completely. In the case when monte carlo $\alpha$ weight is 0 and user $\alpha$ weight is 1 then $\mathcal{D}_{train} = \mathcal{D}_{perc}$ and $\mathcal{D}_{test} = \mathcal{D}_{lin}$, simlarly for the other case we get $\mathcal{D}_{train} = \mathcal{D}_{lin}$ and $\mathcal{D}_{test} = \mathcal{D}_{perc}$. Please note that the distribution $\mathcal{D}_{lin}$ and $\mathcal{D}_{perc}$ have completely different underlying principles and are two completely different distributions.

**Results:** In Figure 3, we show a heatmap plot to which demonstrates the robustness of our method. The color of the block corresponding to Monte Carlo alpha, $\alpha_{mc} = x$ and the users alpha, $\alpha_{user} = y$ represents the fraction of users that were satisfied when $\alpha_{mc} = x$ and $\alpha_{user} = y$. This means that if the user thought of costs only in terms of Linear step involved but the recourse method used samples with only percentile based cost, still the recourse set can satisfy almost the same number of users. In Figure 3, the corners correspond to these extreme cases described above, the user satisfaction for the top left corner ($\mathcal{D}_{train} = \mathcal{D}_{perc}$ and $\mathcal{D}_{test} = \mathcal{D}_{lin}$) is similar to the bottom left corner ($\mathcal{D}_{train} = \mathcal{D}_{lin}$ and $\mathcal{D}_{test} = \mathcal{D}_{lin}$). Similarly things happen for the opposite case which is denoted by the top-right ($\mathcal{D}_{train} = \mathcal{D}_{perc}$ and $\mathcal{D}_{test} = \mathcal{D}_{perc}$) and bottom-right ($\mathcal{D}_{train} = \mathcal{D}_{lin}$ and $\mathcal{D}_{test} = \mathcal{D}_{perc}$) corners. This means that even when a complete distribution shift occurs the performance user satisfaction remains similar. This can be attributed to the hierarchical step for user preference sampling in the procedure because the preferences values can be arbitrary and they scale the raw percentile and linear cost hence the distribution designed this way to model extremely diverse types of transition costs.

This means that **our methods is robust to misspecification in the users' notions of thinking about cost in terms of (relative) percentile shifts or (absolute) linear shifts.** The almost consistent color of the grid means that **there is very slight variation in the Fraction of Satisfied users when the model is tested on out of distribution user cost types.**

**Q6. Does Method Performance Scale with Available Compute?**

**Design:** In this experiment on the Adult-Income dataset, we measure the change in performance of all the models as the number of access to the black-box model (budget) increases. Ideally, a good recourse method should be able to exploit these extra queries and use it to satisfy more users. We vary the allocated budget in the set $\{500, 1000, 2000, 3000, 5000, 10000\}$ and report the FS@1. We run the experiment on a random subset of 100 users for 5 independent runs and then report the average performance with standard deviation-based error bars in Figure 4.

**Results:** In Figure 4, we can see that **as the allocated budget increases the performance of the COLS and P-COLS increases** and then saturates. This suggests that our method can exploit the additional black-box access to improve the performance. Other methods like AR and Face-Knn also

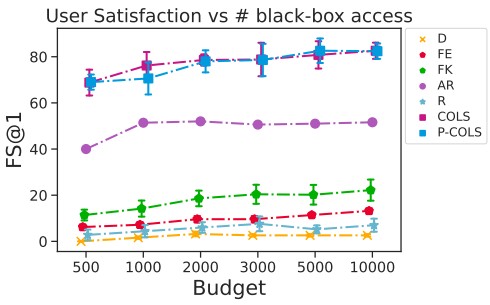
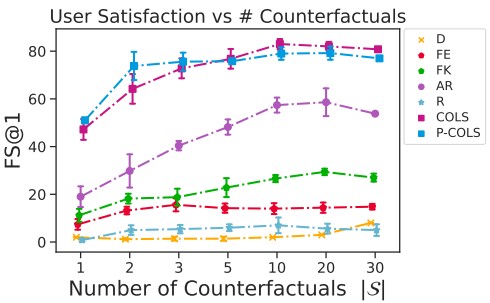

Figure 4: Figure showing the performance of different recourse methods as the Budget is increased. These are the average number across 5 different runs along with the standard deviation error bars. For some methods the standard deviation is very low hence not visible as bars in the plot. It can be seen that as the budget increases the performance of COLS and P-COLS increases. Please refer to Section B.2 for more details.

Figure 5: Figure showing the performance of different recourse methods as the the number of counterfacuals to be generated is increased. These are the average number across 5 different runs along with standard deviation error bars. We see that there is a monotonic increase in the fraction of users satisfied as the size of the set increases. We also observe that most of the performance can be obtained with a small set size. Please refer to Section B.2 for more details.

show performance improvement but our method COLS and P-COLS consistently upper-bound their performance. **Our method satisfies approximately 70% of the user with a small budget of 500** and quickly starts to saturate around a budget of 1000. This suggests that **our methods are suitable even under tight budget constraints** as they can achieve good performance rapidly. For example, in a real-world scenario where the recourse method is deployed and has to cater to a large population, in such cases there might be budget constraints imposed onto the method where achieving good quality solution quickly is required. Lastly, for DICE and Random search the performance on the FS@1 increase by a very small margin and then stays constant as these methods are trying to optimize for different objectives which don't align well with user satisfaction as demonstrated in Section 5.2.

**Q7. Does providing more options to users help?**

**Design:** In this experiment, we measure the effect of having flexibility to provide the user with more options, i.e. a bigger set $\mathcal{S}$. The question here is that can the methods effectively exploit this advantage and provide lower cost solution sets to the user such that the overall user satisfaction is improved. In this experiment on the Adult-Income dataset, we take a random subset of 100 users and fix the budget to 5000, Monte Carlo cost sample is set to 1000 and then vary the size of the set $\mathcal{S}$ in the set $\{1, 2, 3, 5, 10, 20, 30\}$. We restrict the size of the set to a maximum of 30 as beyond a point it becomes hard for users to evaluate all the recourse options and decide which one to act upon. We run 5 independent runs for all the data points and plot the mean performance along with standard deviation error bars. In Figure 5, we plot the fraction of users satisfied @1 as the size of set $\mathcal{S}$ is increased.

**Result:** We observed that **COLS and P-COLS monotonically increase the FS@1 metric as $|\mathcal{S}|$ increases** from 1 to 30. This is consistent with the intuition behind our methods (See Figure 1, section 4.1, B.1.2 for more details). It is a fundamental property of our objective that as $|\mathcal{S}|$ increases towards $M$ which is 1000 in this case, then the quality of the solution set should increase and reach the best possible value that can be provided under the user's cost function. We note empirically that **smaller set size $|\mathcal{S}|$ between 3 to 10 is enough in most practical cases** to reach close to maximum performance. Additionally, even with $|\mathcal{S}| \in \{1, 2, 3\}$ our methods significantly outperform all the other methods in terms of the number of users satisfied. This property is useful in real-world scenarios where the deployed recourse method can provide as little as 3 options while still satisfying a large fraction of users. Additionally, we also see improvement in the case of AR and Face-Knn methods as $|\mathcal{S}|$ increases. Note that Randoms Search's performance doesn't change as we increase the set size because the method doesn't take local steps from the best set and samples random points from a very large space, hence it is much harder to end up with low-cost counterfactuals.

| Feature Name | State Vector | Editable Features | Preference scores | Recourses | Cost |
|---|---|---|---|---|---|
| Age | 24 | No | 0 | | |
| Workclass | Private | No | 0 | (Capital Loss: $0 \rightarrow 1$) | 0.009 |
| Education-Num | 10 | No | 0 | | |
| Martial-Status | Married | No | 0 | | |
| Occupation | Other | Yes | 0.055 | | |
| Relationship | Husband | No | 0 | (Occupation: Other $\rightarrow$ Manager) | 0.378 |
| Race | White | No | 0 | | |
| Gender | Male | No | 0 | | |
| Capital Gain | 0 | No | 0 | | |
| Capital Loss | 0 | Yes | 0.944 | $\left(\begin{array}{l}\text{Occupation: Other} \rightarrow \text{Manager} \\ \text{Capital Loss: } 0 \rightarrow 1\end{array}\right)$ | 0.387 |
| # Work Hours | 40 | No | 0 | | |
| Country | US | No | 0 | | |
| Age | 45 | No | 0 | | |
| Workclass | Private | No | 0 | (Capital Loss: $0 \rightarrow 1$) | 0.071 |
| Education-Num | 7 | Yes | 0.537 | | |
| Martial-Status | Married | No | 0 | | |
| Occupation | Other | No | 0 | (Capital Gain: $0 \rightarrow 1$) | 0.106 |
| Relationship | Non-Husband | No | 0 | | |
| Race | White | No | 0 | | |
| Gender | Female | No | 0 | (Education-Num: $7 \rightarrow 10$) | 0.187 |
| Capital Gain | 0 | Yes | 0.078 | | |
| Capital Loss | 0 | Yes | 0.240 | | |
| # Work Hours | 32 | Yes | 0.142 | (# Work Hours: $32 \rightarrow 70$) | 0.695 |
| Country | US | No | 0 | | |

Table 5: Table providing qualitative examples for two users from the dataset. We show each users state vector, the features that user is willing to edit, the preference scores for those editable features, the recourses provided and the cost of the generated recourses. In the first example we see that user highly prefers the feature *capital loss* and the recourse which suggests edit to that has the lowest cost for the user. Whereas, the recourse which makes changes to both *Occupation* and *Capital Loss* has the highest cost as its changing multiple features. For the second user, we see that the most preferred feature is *Education-Num* but the changes suggested in the recourse requires three steps 7-8-9-10, hence the cost for that recourse is not the lowest but still relatively low. Whereas, the recourse suggesting smaller changes to *Capital Loss* which is the second most preferred feature has the lowest cost for the user.

**Q8. Does increase the number of Monte Carlo samples help with user satisfaction?**

**Design:** In this experiment, we want to demonstrate the effect of increasing the number of Monte Carlo samples on the performance of our COLS method. We take a random subset of 100 users, a budget of 5000, $|\mathcal{S}| = 10$. We vary the number of Monte Carlo samples (M) in the set $\{1, 5, 10, 20, 30, 100, 200, 300, 500, 1000\}$ and compute the user satisfaction. We ran 5 different runs with different Monte Carlo samples and show the average FS@1 along with the standard deviation in the Figure 6.

**Results:** We observe that **as the number of Monte Carlo samples increases, the performance of the method on the FS@1 metric monotonically increases.** This supports the intuition underlying our method (see Figure 1). That is, given a user with a cost function $\mathcal{C}_u$ as we get more and more samples from the cost distribution $\mathcal{D}_u$ the probability of having a cost sample similar to $\mathcal{C}_u$ increases and hence the fraction of satisfied users increase. It is impor-

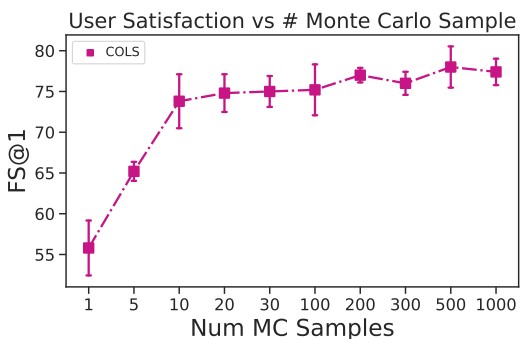

Figure 6: Figure showing the performance of the COLS method as the number of Monte Carlo samples increase. These are the average number across 5 different runs along with standard deviation error bars. There is a steep increase and then the performance saturates. This implies that in practice we do not need a large number of samples to converge to the higher user satisfaction. Refer to Section B.2 for more details.

tant to note that **empirically the method's performance approaches maximum user satisfaction with as low as 20 Monte Carlo samples.** In real-world scenarios, where the deployed model is catering to a large population this can lead to small recourse generation time, hence making it more practical.

| Data | Method | Metrics | | | | | | |
|---|---|---|---|---|---|---|---|---|
| | | Cost Metrics | | | Distance Metrics | | | |
| | | FS@1 | PAC(↓) | Cov | Div | Prox | Spars | Val |
| **Adult-Income - NB** | **DICE** | 6.28 | 1.45 | 27.01 | 53.01 | 57.02 | 47.80 | 86.20 |
| | **Random** | 0.08 | 2.42 | 17.41 | **70.35** | 33.32 | 22.45 | 75.71 |
| | **COLS** | 72.67 | **0.36** | **74.60** | 29.27 | **79.06** | **76.64** | **97.85** |
| | **P-COLS** | **70.03** | 0.39 | 72.81 | 29.85 | 78.45 | 76.29 | 92.30 |
| **COMPAS - NB** | **DICE** | 14.86 | 1.02 | 25.45 | 27.88 | 82.38 | 69.44 | **99.86** |
| | **Random** | 1.31 | 1.87 | 21.76 | **49.07** | 54.10 | 42.34 | 67.82 |
| | **COLS** | 67.34 | **0.31** | 68.11 | 20.53 | 85.47 | 82.34 | 95.97 |
| | **P-COLS** | **70.86** | 0.35 | **72.03** | 21.03 | **85.48** | **82.88** | 91.93 |

Table 6: Table comparing different recourse methods across various cost and distance metrics on Non-Binary versions of the datasets (Section B.1.1).The numbers reported are averaged across 5 different runs. Variance values have been as 89% of them were lower than 0.05, with the maximum being 0.86. FS@1: Fraction of users satisfied at $k = 1$. PAC: Population Average Cost. Cov: Population Coverage. For all the metrics higher is better except for PAC where lower is better.

**Q9. Qualitative examples of the recourses generated for some of the users.**

In Table 5, we show a few examples of users along with their state vector, their editable features, their preference scores along with the recourses provided to them and their cost.

**Q10. Comparison of methods on Non Binary Dataset?**

In Table 6, we show the results on the non-binary version of the dataset. We observe similar performance on and trends in these results as well. COLS and P-COLS performs the best in terms of user satisfaction.

---

**Algorithm 4:** Sampling procedure for Percentile Cost Mean

---

1 **function** getPercentileCostMean($\boldsymbol{s}, p^{(f_i)}, f_i, \mathcal{F}_p$)
      // $s_i$ value of feature $f_i$ in s.

2   **if** $f_i \notin \mathcal{F}_p$ **then**

3      $\mu^{(f_i)}(s_i, .) = \infty$

4      $\mu^{(f_i)}(s_i, s_i) = 0$

5   **else**

6     **if** $f_i$ *is ordered* **then**

7       **if** $f_i$ *can only increase* **then**

8         
$$\mu^{(f_i)}(s_i, x) = \begin{cases} |getPercentile(x) - getPercentile(s_i)| & \forall x > s_i \\ 0 & \forall x = s_i \\ \infty & \forall x < s_i \end{cases}$$

9       **else if** $f_i$ *can only decrease* **then**

10         
$$\mu^{(f_i)}(s_i, x) = \begin{cases} |getPercentile(s_i) - getPercentile(x)| & \forall x < s_i \\ 0 & \forall x = s_i \\ \infty & \forall x > s_i \end{cases}$$

11       **else if** $f_i$ *can both increase or decrease* **then**

12         
$$\mu^{(f_i)}(s_i, x) = \begin{cases} |getPercentile(x) - getPercentile(s_i)| & \forall x > s_i \\ 0 & \forall x = s_i \\ |getPercentile(s_i) - getPercentile(x)| & \forall x < s_i \end{cases}$$

13     **else if** $f_i$ *is unordered* **then**

14       $\mu^{(f_i)}(s_i, .) = Uniform(0, 1)$

15   $\mu^{(f_i)}(s_i, .) \leftarrow \mu^{(f_i)}(s_i, .) * (1 - p^{(f_i)})$

16   $\sigma^{(f_i)}(s_i, .) \leftarrow 0.01$

17   **return** $\mu^{(f_i)}$, $\sigma^{(f_i)}$

---

---

**Algorithm 5:** Sampling procedure for Linear Cost Means

---

1 **function** getLinearCostMean($s, p^{(f_i)}, f_i, \mathcal{F}_p$)

2     **if** $f_i \notin \mathcal{F}_p$ **then**

3         $\mu^{(f_i)}(s_i, .) = \infty$

4         $\mu^{(f_i)}(s_i, s_i) = 0$

5     **else**

6         **if** $f_i$ *is ordered* **then**

7             **if** $f_i$ *can only increase* **then**

8                 $\mu^{(f_i)}(s_i, x) = \begin{cases} \frac{|\{y \ | \ y > s_i \wedge y \leq x\}|}{|\{y \ | \ y > s_i\}|} & \forall x > s_i \\ 0 & \forall x = s_i \\ \infty & \forall x < s_i \end{cases}$

9             **else if** $f_i$ *can only decrease* **then**

10                 $\mu^{(f_i)}(s_i, x) = \begin{cases} \frac{|\{y \ | \ y < s_i \wedge y \geq x\}|}{|\{y \ | \ y < s_i\}|} & \forall x < s_i \\ 0 & \forall x = s_i \\ \infty & \forall x > s_i \end{cases}$

11             **else if** $f_i$ *can both increase or decrease* **then**

12                 $\mu^{(f_i)}(s_i, x) = \begin{cases} \frac{|\{y \ | \ y > s_i \wedge y \leq x\}|}{|\{y \ | \ y > s_i\}|} & \forall x > s_i \\ 0 & \forall x = s_i \\ \frac{|\{y \ | \ y < s_i \wedge y \geq x\}|}{|\{y \ | \ y < s_i\}|} & \forall x < s_i \end{cases}$

13         **else if** $f_i$ *is unordered* **then**

14             $\mu^{(f_i)}(s_i, .) = Uniform(0, 1)$

15     $\mu^{(f_i)}(s_i, .) \leftarrow \mu^{(f_i)}(s_i, .) * (1 - p^{(f_i)})$

16     $\sigma^{(f_i)}(s_i, .) \leftarrow 0.01$

17     **return** $\mu^{(f_i)}, \sigma^{(f_i)}$

---

