# OpenReview forum: "Low-Cost Algorithmic Recourse for Users With Uncertain Cost Functions"
_ICLR.cc/2022/Conference — ICLR 2022 Submitted_

### Official Review · Reviewer_B3sM · 2021-11-02

**Correctness:** 2
**Technical Novelty And Significance:** 3
**Empirical Novelty And Significance:** 3
**Recommendation:** 3
**Confidence:** 3

**Main Review:**

I do not think this papers achieves what it sets out to do. It is mentioned in the related work, how the closest literature to the paper is other cost-based approaches to finding recourse and different from those approaches, this paper drops the assumption that there is a known global cost function that is shared by all users. First, the paper formulates $\mathrm{MinCost}(\cdot;\mathcal{C}_u)$ as the cost function of user $u$, which is characterized by the unknown transition matrices $\mathcal{C}_u$. However then, the paper assumes a distribution $\mathcal{D}$ over $\mathcal{C}_u$ 's of different users is known and proposes to optimize $\mathbb{E}_\mathcal{C}{}_\sim{}_\mathcal{D}[\mathrm{MinCost}(\cdot;\mathcal{C})]$, which is effectively a known global cost function (with respect to user state $\mathbf{s}_u$ and recourse set $\mathcal{S}$). Is this not the case?

Having said that, the proposed cost function has a certain structure and it is still novel: (i) authors propose a hierarchical cost distribution as the particular $\mathcal{D}$ they consider and (ii) by only considering the element with the minimum cost for each sample from $\mathcal{D}$, they exploit the fact that each users only really requires one recourse that they are happy with to be satisfied. Proposing a new cost-based objective like this could still be a valuable contribution. But then, the paper needs to be positioned accordingly and highlight the merits of optimizing a cost-based objective structured in this new way. Note that the current experiments are not helpful in comparing against other cost-based objectives proposed in previous work: cost functions of the users are simulated according to the proposed cost function, then of course, a method that optimizes it would perform better than methods optimizing other cost functions.

Some of the conclusions made in the results section suffer from user preferences being simulated as well. For instance, at the end of "Q2," the authors conclude that high diversity is not necessary to satisfy individual users; this is of course true for the simulated users since their cost function is designed to ignore diversity in the first place.


**Summary Of The Paper:**

This paper aims to find algorithmic recourse that has low-cost to the users. Unlike previous work, the authors do not assume that there is a known global cost function that is shared by all users.

**Summary Of The Review:**

I believe the claim that the paper relaxes the assumption of knowing a global cost function is not true. However, it still introduces an interesting new objective to optimize for when finding algorithmic recourse.

---

> ### Author Response · Authors · 2021-11-14
>
> We thank you for your time, effort, and comments. We have also updated the paper to disentangle the notations a bit, kindly have a look at the general comment and the paper. We are looking forward to the discussion!
>
> **R4.1 - Concerns regarding known global cost function distribution.**
>
> We have added the clarification to this in the general comment on “ **5. Approximations in EMC** ”, and “ **1. Is this new evaluation procedure any good?** ”.
>
> **R4.2 - Circularity of evaluation due to simulated user cost functions.**
>
> We have added the clarification regarding these points in the general comment “ **2. Is the evaluation of COLS fair, given that it uses the same distribution at train and test time?** ", and “ **3. What happens if the train and test time cost function distributions differ?** ”. Kindly have a look at it and leave us more questions if you have any!
>
> **R4.3 - Some of the conclusions made in the results section suffer from user preferences being simulated as well. For instance, at the end of "Q2," the authors conclude that high diversity is not necessary to satisfy individual users; this is of course true for the simulated users since their cost function is designed to ignore diversity in the first place.**
>
> We first refer you to the general comments “ **4. Is it fair to compare COLS vs other methods, since COLS uses $D_{test}$ at train time?** ” and “ **1. Is our evaluation procedure realistic or preferable to past evaluations?** ”
>
> We view it as a striking result that optimizing for diversity of counterfactuals is not the best way to satisfy a diverse set of people. Here we argue why the conclusions at the end of Q2 about diversity holds. If diversity correlates well with user satisfaction. This implies that no matter what the user’s hidden cost function is, providing a diverse solution set should increase the likelihood that a user will be provided with at least one recourse with a cost less than the threshold k. This implies that given enough users, the fraction of users satisfied will be more if the solutions were diverse as opposed when they are not, irrespective of the method used to simulate cost functions. If we look at the methods with high diversity in Table-1 and Table-5 we see that having high diversity doesn’t lead to lower population average cost or higher user satisfaction. Furthermore, Act. Recourse, Face and COLS and P-COLS achieve higher user satisfaction even when the diversity is not as high as that of Random. Note that in our experiment we have approximately 750 users for the adult dataset and we ran 5 independent runs and the results are average across them.
>
> We are not sure what it means that the “cost function is designed to ignore diversity”. Diversity makes sense in case of cost function sampling distribution D. There might be distributions which inhibit diversity in types of cost functions they can model. This is opposite to what we are doing, our proposed distributions $D_{mix}$ are designed to be flexible. It can capture all possible feature subsets which users might consider preferable along with all combinations of linear/percentile costs (see section 4.2 and Appendix A.2.1). Hence, the cost function samples from them are diverse.

---

### Official Review · Reviewer_Fy3U · 2021-11-03

**Correctness:** 3
**Technical Novelty And Significance:** 3
**Empirical Novelty And Significance:** 2
**Recommendation:** 5
**Confidence:** 3

**Main Review:**

I enjoyed reading this paper in general. My major comments are:

1.	In Section 4.1, it is assumed that there is a distribution over all the cost functions D_c for the population. Is the distribution D_c known or unknown? A more practical setting is that D_c is unknown. Then how to use Monte Carlo Estimation to approximate the expectation of the MinCost?

For different users u, it is assumed that C_u follows distribution D_c. However, in the introduction, it claims that “we propose a method for identifying a user-specific recourse set that contain at least one good solution for the user”. However, it seems inconsistent between the motivation and the assumption. Why do all users share the same distribution of the cost function? Is the framework generalizable to the setting with different distribution?

2.	Theorem 4.1 proves the monotonicity of Cost-Optimized Local Search Algorithm. But how does ExpMinCost(s_u, S_t^best; {C_i}_{i=1}^M) converge? Theorem 4.1 does not imply that, but it is a very important question.

3.	Why choosing Equation (3) and Equation (4) as metrics to measure recourse quality? What is the advantage of choosing a threshold function? How to choose k in real cases?

4.	In the numerical experiments, could you compare with other functions that measure the recourse quality in previous recourse papers? I think my main concern is on FS@k. Is using FS@k equivalent to the following: assume there exists a black-box algorithm that can output the indicator that if the total cost is smaller than k, then the distance function can be used to measure the recourse quality.
 Could you use numerical experiments to emphasize the advantage of using FS@k compare to other measure functions such as weighted sum of costs?

5.   What is the computational complexity of your algorithm? How is that compared to other benchmarks?

6.    Why is fairness an important issue in this work? Could you comment more on this part to motivate?


**Summary Of The Paper:**

This work introduces a new method for identifying actionable recourses for users with user-specific cost functions. Users’ cost functions are hidden from the recourse method. The paper proposed a discrete optimization algorithm COLS to solve the objective EMC. It further used a popular real-world dataset to illustrate the performance.

**Summary Of The Review:**

This paper studied an interesting problem. To improve the paper, the author may want to illustrate the advantage of using FS@k and how it is very different from state-of-art measure functions, both conceptually and numerically.

---

> ### Author Response · Authors · 2021-11-14
> **Part [2/2]**
>
> **R3.7 - I think my main concern is on FS@k. Is using FS@k equivalent to the following: assume there exists a black-box algorithm that can output the indicator that if the total cost is smaller than k, then the distance function can be used to measure the recourse quality.**
>
> Thank you for the detailed question. We are not completely sure of our understanding of this concern. Our current interpretation of this point is: Assuming we have access to an oracle which takes in a user state along with a recourse and tells us if the user cost under their cost function is less than k or not. Then if we have multiple recourse options for which the oracle tells us that the cost is under k, then will the recourse with the minimum distance also be the one with lowest cost?
>
> If our understanding of the question is correct then the answer is No, because a recourse with minimum distance with the user's current state might not have the lowest cost if those edited features are not the most preferred features by the user. For example, if there are two features A and B, and say A is preferred over B then bigger changes can be made in A  while still keeping the cost low. Whereas small changes in the feature B can lead to higher costs. Hence, it depends on the user’s preference and cost function, and distance might not be enough to find lowest cost recourse even in such cases.
>
> **R.3.8 - Advantage of FS@k and numerical comparison with weighted averages of costs.**
>
> In our experiments, we also report the population average cost (PAC) which is the average cost incurred by the users who are provided with valid recourses. Metrics like PAC which are a weighted average make it harder to measure individual user’s satisfaction. For example, say all men have recourse costs much less than k and all females have recourse costs higher than k then PAC can still be low, say below k. In this case, the average cost is low but many users might obtain solutions with much higher costs, hence satisfiability thresholding has to be done before aggregating over users.
>
> **R3.9 - Computational Complexity and comparison to others.**
>
> Please refer to the general comment on “**6. Computational complexity and Runtimes**”.
>
> **R3.10 - Importance of fairness.**
>
> Fairness is an important ethical consideration in machine learning generally, and it is relevant here because while recourse methods can be used to provide value to users, we would have ethical concerns if that value was distributed unfairly. For instance, it would be an ethical concern if only men were given actionable recourse to get financial loans approved, while women were never given actionable recourse. Fairness is important to recourse, above and beyond the fairness of the underlying decision-making system: it could be that men and women are approved for loans equally often in the “first round” of a decision-making system, but if men’s petitions are approved more often than women’s, the result would still be an unfair system. Ensuring that recourse methods provide fair solutions is an important step in combating potential inequities such as this.
>
> [1] Ramaravind K Mothilal, Amit Sharma, and Chenhao Tan.  Explaining machine learning classifiers through diverse counterfactual explanations. FAT 2020
>
> [2] Kaivalya Rawal and Himabindu Lakkaraju.  Beyond individualized recourse:  Interpretable and interactive summaries of actionable recourses. NeurIPS 2020

---

> > ### Comment · Reviewer_Fy3U · 2021-11-23
> > **Reply to the authors**
> >
> > Thanks for the response. It has addressed concerns that I had in the initial review report.

---

> ### Author Response · Authors · 2021-11-14
> **Part [1/2]**
>
> We thank you for your time, effort, and comments. We have also updated the paper to disentangle the notations a bit, kindly have a look at the general comment and the paper. We are looking forward to the discussion!
>
> **R3.1 - Is the distribution D_c known or unknown and how Monte Carlo works when D_c is unknown.**
>
> We have updated the paper to disentangle the notation a bit and refer you to section 4.1. There is a true cost function sampling distribution $D^*$ which is unknown to us but we use another distribution $D_{train}$ with EMC to induce diversity with respect to cost function in the recourse set. We request you to look at the general comment " **5. Approximations in EMC** ".
>
> **R3.2 - For different users u, it is assumed that C_u follows distribution D_c. However, in the introduction, it claims that “we propose a method for identifying a user-specific recourse set that contain at least one good solution for the user”. However, it seems inconsistent between the motivation and the assumption. Why do all users share the same distribution of the cost function? Is the framework generalizable to the setting with different distribution?**
>
> In the paper, the assumption is that the true population cost function distribution $D^*$ is common to all the users but the user cost functions which are sampled from $D^*$ are different from each other. Given the distribution $D^*$ is flexible enough to model various types of user preferences and cost functions, users don’t require their own individual distributions. We provide an example of one such flexible distribution $D_{mix}$ which can give rise to very diverse cost functions in terms of user preferences, please refer to Section 4.2 to know more about the flexibility of the distribution. Additionally, we would like to note that, it is easy to extend our objective EMC to include cost functions from different plausible distributions, $D^1, ... , D^m$. This is equivalent to a setting where each cost function is drawn from a mixture model which is a mixture of plausible distributions $D^1, ... , D^m$.
>
> **R3.3 - Convergence of COLS**
>
> Thanks for this great question! We first note that $ExpMinCost(s_u, S^{best}_{t}; \{C_i\}_{i=1}^M)$ is bounded below by 0, i.e. the EMC >= 0 always as we cannot have cost lower than zero. Next, we invoke results from real analysis, that given a sequence which is monotonically decreasing and bounded below converges. Hence, the convergence of COLS with EMC is also guaranteed. We empirically tested for the convergence behavior in Appendix Fig 4, as we let different recourse methods run for longer. We observe that the COLS + EMC converges with a budget as low as 1k.
>
>
> **R3.4 - Why choosing Equation (3) and Equation (4) as metrics to measure recourse quality? What is the advantage of choosing a threshold function?**
>
> The equations (3) and (4) follow directly from past work [2] in recourse methods, where two goals are to provide low-cost solutions to users and to provide solutions to a large fraction of users. We choose a threshold function to convert the user cost into our FS@k metric because our cost is unitless, which would make the reported Population average cost (PAC) somewhat difficult to understand. In comparison, we have a stronger intuitive understanding of what FS@k means, by analogy, it’s like reporting accuracy rather than test loss. Another intuition is that recourse is worthwhile to any user when the cost is under a threshold k because below a certain cost the marginal utility of selecting one recourse over the other is small. In our experiments, we keep k fixed across users and tasks because the goal of any recourse method is to find low-cost recourse set regardless of the threshold.
>
> **R3.5 - How to choose k in real cases?**
>
> In the real world setting a small survey can be done where the users are provided with recourses and asked to mark all acceptable recourses. Given that we know the cost for all recourses which are accepted by the user this can be used to determine the threshold.
>
> **R3.6 - Comparison with metrics from past work which measure recourse quality?**
>
> We would like to clarify that we do compare the same set of solutions with other metrics used in the past works, specifically diversity [1], proximity [1], sparsity [1], validity [1] and coverage [2].

---

### Official Review · Reviewer_cyuv · 2021-11-04

**Correctness:** 3
**Technical Novelty And Significance:** 4
**Empirical Novelty And Significance:** 4
**Recommendation:** 6
**Confidence:** 2

**Main Review:**

Pros
-	This paper proposes a new way of evaluating user satisfaction which differs from existing methods that measures on heuristics such as distance/diversity or assume a fixed global user cost function. It is more flexible and realistic, and thus could be an interesting direction to follow. The proposed formulation is quite novel and technically non-trivial, with some theoretical grounding.
-	The experimental results are very strong on the 3 newly proposed metrics. The authors also conduct extensive ablation studies on different aspects of the problem, although many of them are deferred to the supplementary.
-	The discussion is pretty comprehensive, and they also included a fairness analysis.

Concerns
-	One major issue is on the readability of the paper. It is certainly good that the paper contains a lot of information, however currently it seems that the main text is a bit too packed such that very limited detail about the main methodology is provided. In fact, both the core sampling and optimization algorithms are described in the appendix, and it is very hard for this reviewer to understand them solely based on the descriptions in section 4. Perhaps the authors could reorganize the content such that less space is spent on repeating the contributions/motivations.
-	As algorithmic recourse is a rarely new domain and may not be well-known by general audience, it might be better to translate the domain-specific terminologies into plain language or more general language in ML in the introduction part.
-	It seems the newly introduced evaluation metrics are generated using the same sampling distribution used for computing EMC, wouldn’t that be a bit circulated to evaluate something where the ground-truth is closely related to the objective used for optimization? Is there any way to evaluate on more realistic user cost rather than simulating it with the same distribution as the one used in EMC? The authors talk about distributional shift regime in the appendix, that still the ground-truth distribution is from the same family of the EMC distribution (mixture of percentile shift and linear cost). It might be more convincing if it is from a totally independent distribution.

Questions
-	What is used as the initial starting set for COLS?
-	In the problem formulation in (2), does it mean that the best recourse set would consist at least one desired outcome solution but it may not be the one with lowest cost? If so how do one achieve balance between outcome and satisfaction?
-	What is the computational complexity of the algorithm?
-  Is there any downside from the underperformance in distance-based metrics?


**Summary Of The Paper:**

In this paper, the problem of algorithmic recourse is studied where the goal is to find best recourse (counterfactual set) that is optimized for user cost. The author proposed new user-incurred cost evaluation method, Expected Minimum Cost (EMC), which approximate user satisfaction without assuming a fixed global user cost function, and instead consider user cost functions as hidden and user-specific. Specifically, the authors define cost function for each user as a set of feature-specific functions of user-incurred cost when transitioning between feature states, and define MinCost as the minimum transition cost across possible recourses. To cover diverse user cost functions, they propose to model user cost distribution with a hierarchical sampling procedure and estimated expected minimum cost by drawing samples from it. Next, they formulate a discrete optimization problem using EMC as objective, and propose a search algorithm (COLS) for best recourse generation. They introduce three new metrics for user satisfaction (all related to MinCost): FS@k, Coverage and PAC. Finally, they test with two real-world datasets and show that COLS achieves significant outperformance against baselines (that optimize for distance-based metrics) on the newly proposed metrics and show that their method is doing better in fairness as well.

**Summary Of The Review:**

In this paper the author proposed a new way for evaluating and optimizing user satisfaction. The technical contributions are solid and the results are rather promising despite the potential bias toward EMC. The paper contains fruitful discussion and ablation studies, although it can be further improved in terms of clarity. Therefore, I would like to give a weak accept.

---

> ### Author Response · Authors · 2021-11-14
>
> We thank you for your time, effort, and comments. We have also updated the paper to disentangle the notations a bit, kindly have a look at the general comment and the paper. We are looking forward to the discussion!
>
> **R2.1 - Readability of the paper.**
>
> Thank you very much for the suggestion. We will further update the paper in a few days to clarify the methodology in the main paper.
>
> **R2.2 - Circularity in the evaluation due to using the same distribution used for EMC and simulating cost functions.**
>
> This is a great question! We have provided clarification to this in the general comment “ **4. Is it fair to compare COLS vs other methods, since COLS uses $D_{test}$ at train time?** ”, and “ **1. Is our evaluation procedure realistic or preferable to past evaluations?** ”.
>
> **R2.3 - Distribution shift on a more realistic and unrelated distribution.**
>
> We point to the general comment “ **3. What happens if the train and test time cost function distributions differ?** ”. Note that, $D_{mix}$ is a mixture of two independent and disjoint distribution $D_{perc}$ and $D_{lin}$ and the extreme cases of mixing weight correspond to them, i.e $D_{train}$ and $D_{test}$ have completely different underlying phenomena defining transition costs. We tried to develop other settings where the distribution differs completely in some other manner but trying to do so led to non-intuitive properties like, should cost functions be non-monotonic? Should cost functions not relate to either percentile shift or linear steps? In general, we realised that creating distribution by violating these assumptions leads to non-intuitive cost functions.  If you have any suggestions you want us to try please let us know we would be happy to report numbers on them.
>
> **R2.4 - What is used as the initial starting set for COLS?**
>
> We take the user’s state vector s_u and perturb the features randomly with a Hamming distance of two in order. We repeat this $|S|$ number of times to get the initial set $S$.
>
> **R2.5 - The best recourse set would consist at least one desired outcome solution but it may not be the one with lowest cost?**
>
> This is a great question! As noted in Appendix B.1.2, the best recourse set might not contain the lowest cost solution for that user's simulated cost function because of multiple reasons, 1) We run the algorithm with a constraint on the budget. 2) The EMC objective is trying to find a solution set which works reasonably well for all the sampled cost functions used in EMC, hence it favors recourses which have low cost for multiple sampled cost functions, 3) the size of the set $S$ is restricted. Note that, we don’t necessarily need to provide the user with the lowest cost solution, we need to provide them with one solution for which the cost is low.
>
> **R2.6 - If so how do one achieve balance between outcome and satisfaction?**
>
> We are not sure if we understand the question correctly. But In our case, the FS metric’s threshold k is what characterizes low cost and user satisfaction. All recourses with costs below a certain threshold are acceptable to the user. Another intuition is that recourse is worthwhile to any user when the cost is under a threshold k because below a certain cost the marginal utility of selecting one recourse over the other is small. In our experiments, we keep k fixed across users and tasks because the goal of any recourse method is to find low-cost recourse set regardless of the threshold.
>
> **R2.7 - What is the computational complexity of the algorithm?**
>
> Please refer to the general comment on “ **6. Computational complexity and Runtimes** ”.
>
> **R2.8 - Is there any downside from the underperformance in distance-based metrics?**
>
> There are several issues with distance-based metrics which make their use unimportant cost-based formulations. 1) The distance metrics try to approximate user satisfaction indirectly. 2) None of them individually provide a complete picture of whether a user will be satisfied or not. 3) These metrics are on different scales and it's hard to determine what is a reasonable way to merge them such that the result correlates user’s satisfaction. 4) Even after normalizing them to the same scale the issue is that all of these metrics have different units and some of them are unitless. If the true user cost functions are known for evaluation purposes then it overrides all distance-based metrics and provides a more direct way which resolves all the issues mentioned in points 1, 2, 3, 4 above.

---

> > ### Comment · Reviewer_cyuv · 2021-12-02
> > **reponse to the rebuttal**
> >
> > I appreciate the authors' explanation, however the major concern here is not that train/test are coming from same distribution, but rather the EMC method directly uses a model family that is the same from (or covering) the underlining train/test data distribution, which is not the same case as in supervised learning. Given the current results, it is still not fully convincing whether this would work on realistic data. Thus I would remain my current score (weak accept).

---

> > > ### Author Response · Authors · 2021-12-02
> > > **Response to Reviewer cyuv Comment**
> > >
> > > Thanks for the comment. We are not quite sure if we understand your current concern, and it would be greatly appreciated if you can elaborate on it a bit further. We are listing down a few things we are not clear about.
> > >
> > > - What do you mean by the model family? Do you mean the distribution used to sample cost functions for EMC? If yes, then that is the $D_{train}$ distribution as mentioned in Equation 5 and Updated Notation in general response 1.
> > >
> > > - What do you mean by the model family covering the train/test time distribution?
> > > The train distribution is the one used for optimizing EMC and the test distribution is the one used for sampling hidden cost functions for experiments to evaluate the incurred cost. We address the questions about the train and test distribution in general response 2, 3, and 5; please also see the updated notation for better clarity. If there was something about these responses which you feel is unsatisfactory, it would be really helpful if you can point to those parts so we can address your concern.
> > >
> > > Regarding your ultimate concern with the applicability of our method to realistic data, we feel that we demonstrate that our method would be useful in a realistic setting due two new aspects of our work: (1) We use an expressive cost function distribution which allows us to model individual users with diverse and plausible cost function samples as opposed to past works where assume all users have the same cost function which makes our method much more realistic. (2) We show the robustness of our method to strong distribution shifts in cost function distribution, for example using a percentile cost distribution as $D_{train}$ to optimize for EMC and using a linear cost distribution as D_{test} to sample hidden user cost functions (and vice-versa).

---

### Official Review · Reviewer_1gBz · 2021-11-07

**Correctness:** 2
**Technical Novelty And Significance:** 2
**Empirical Novelty And Significance:** 2
**Recommendation:** 3
**Confidence:** 3

**Main Review:**

The paper addresses an important issue of dealing with model bias in producing fair user outcomes. My main concern with the paper is the lack of clear supporting arguments on why the choice of cost models (including knowing the distribution over the cost functions) is the right one? I am not clear about the assumptions made in the proposed cost function that each user adopts.  First, how is the cost function effectively computed even with the use of a recourse set? Next, how is the recourse set itself guaranteed to always produce at least one reasonable counterfactual in the set?  More specifically, even as the authors acknowledge knowing the exact cost function by each user is difficult, their explanation of using the recourse set to get around this problem with high confidence is unclear to me.  Intuitively, it seems a measure like diversity will be more effective when the cost functions are unknown/private to the user.


**Summary Of The Paper:**

The authors propose a recourse methodology to deal with model biases/fairness issues in producing equitable outcomes for all users (classes).

**Summary Of The Review:**

I am not completely convinced the proposed model of computing a recourse set to minimize the expected cost for the user is always effective in the absence of knowing the cost function even approximately.   The experiments to show the effectiveness with respect to the proposed baseline are inconclusive with respect to natural measures like diversity.

---

> ### Author Response · Authors · 2021-11-14
>
> We thank you for your time, effort, and comments. We have also updated the paper to disentangle the notations a bit, kindly have a look at the general comment and the paper. We are looking forward to the discussion!
>
>
> **R1.1 - Lack of clear supporting arguments, choice of cost models and assumptions in cost functions.**
>
> In the paper, we don’t assume that the users' true cost functions are known, although we do use the same $D_{train}$ and $D_{test}$ in some experiments. Note that we establish that our method is robust to plausible distribution shifts (see the general comment “ **2. Is the evaluation of COLS fair, given that it uses the same distribution at train and test time?** ”, “ **3. What happens if the train and test time cost function distributions differ?** ”). For our approximate cost distribution D_{mix}$, we propose a plausible distribution which we use to induce diversity in the generated recourse set with respect to various cost functions. The only major assumption made to create the plausible distribution is that users think of cost either as percentile shift or linear steps involved in the transition. We refer you to the general comment “ **1. Is our evaluation procedure realistic or preferable to past evaluations?** ” and to section 4.2 and Appendix A 2.1.
>
> **R1.2 - How is the cost function effectively computed?**
>
> Given a cost function and a recourse set we use Equation 1 described in the paper to compute the cost. Equation 1 essentially computes the cost of all the recourses in the set and then outputs the minimum cost. Note that the cost function $C^*_u$ in Equation 1 can be replaced with any other cost function to compute cost with respect to it.
>
> **R1.3 - How is the recourse set itself guaranteed to always produce at least one reasonable counterfactual in the set?**
>
> We would like to clarify that we do not claim COLS is guaranteed to produce a set with at least one reasonable counterfactual. Rather, COLS when optimizing for EMC tries to find a set of counterfactuals which works well across the sampled cost functions used in EMC, and we show empirically that COLS succeeds at this (see Table 1, Table 5). In our paper, we do claim that the COLS optimization method guarantees a monotonic reduction in the EMC objective, a proof of which can be seen in Appendix A.2.2.
>
> **R1.4 - More specifically, even as the authors acknowledge knowing the exact cost function by each user is difficult, their explanation of using the recourse set to get around this problem with high confidence is unclear to me.**
>
> We refer you to the general comment on “**5. Approximations in EMC**”.
>
> **R1.5 - Intuitively, it seems a measure like diversity will be more effective when the cost functions are unknown/private to the user.**
>
> We agree with you that diversity (as defined in Sec 5.1 - Metrics) could be one good way to provide a user with a low-cost solution but it is not guaranteed to do this. In scenarios where methods adopt the direct objective of modeling user incurred cost, all the distance-based metrics are irrelevant because cost is the ultimate metric that quantifies how easy it is for the user to actually adopt the recourse. In our experiments (see Table 1: Random row, Table2: Diversity row, Table 6: Dice and Random rows), we find that these methods promote diversity but don’t lead to higher user satisfaction. In comparison, optimizing for diverse cost functions is much more likely to provide a user from a diverse population with a good solution as shown by the result on COLS and P-COLS.

---

### Author Response · Authors · 2021-11-14
**General Response [2/2]**

4. **Is it fair to compare COLS vs other methods since COLS uses $D_{test}$ at train time?**

At first glance, we agree one might think that it is trivially true that a method that optimizes for one cost distribution will do better than other methods that do not optimize for that distribution. But ultimately we do not agree that the comparison is not helpful, for the following reason: We suggest that our evaluation procedure is simply more realistic than that in past work. It is more realistic that people would have a variety of cost functions with plausible properties, than a single shared cost function. If this is true, then whenever method A outperforms method B on this evaluation, the details of how method A vs method B works do not undermine the main conclusion, which is that method A works better than method B in a realistic evaluation setting. Given this conclusion, we can then attribute the difference in outcomes to the differences in the methods, i.e. we can make valid claims like “method A works better than method B in this more realistic evaluation because the difference between the methods is their objective.”

5. **Approximations in EMC**

As mentioned in Section 4.1, the EMC objective requires three things, the user's current state $s_u$, a recourse set $S$, and the cost samples $C_i \sim D_{train}, \forall i \in [M]$. We make two approximations, the first approximation is that $D_{train}$ might not be the same as the population’s true cost function distribution $D^*$ and the second while doing Monte Carlo estimation to compute the expectation. In Appendix B.2 Q8 and Figure 6, we show that even small M works well in practice. As noted in Section 4.1 Remark, EMC’s solution set is expected to have low cost with respect to cost function samples from $D_{train}$. If this $D_{train}$ is general enough to capture diverse cost functions then we can expect the generated recourse set to work well on the test distribution $D_{test}$ as well.

6. **Computational complexity and Runtimes.**

COLS is a local search-based method and runs for $O(\frac{B}{|S|})$ iterations for each user to generate the recourse set, where B is the budget (see section 5.1 - Baselines). The complexity of the cost optimization step in COLS is $O({|S|}^2 * M)$ per iteration. Values of $|S|$ and M as low as 3 and 10 respectively work well in practice (see Appendix B.2 Q7, Q8 and Figure 5, 6). Finally for the current implementation the wall clock time on the adult dataset for each user with $|S|$ = 10, M = 100, B = 5000 setting is COLS = 20s, Random = 7.5s, DICE = 7.5s, AR = 11s, Face-knn = 7s, Face-Eps = 6s (can be parallelized across users). Cost function samples can be pre-computed once and saved for all experiments, this typically takes a few minutes (&lt;5 min) across all users.

---

### Author Response · Authors · 2021-11-14
**General Response [1/2]**

### **Updates to the paper**

We have updated the paper to make some changes to improve the clarity of the paper



* We have updated the notation to improve clarity regarding various distributions and cost functions in Sections 4.1 and 4.2.
* We have updated the notation for defining a user in Section 3. The users' true cost functions are denoted by $C^*_u$ and population true distribution by $D^*$.
* Updated the description of simulated cost functions in the experimental setup (Section 5.1).
* We have updated the description of robustness experiments to clarify their settings and implications in Appendix Section B2, Q5.

**Updated Notation**

Users have a hidden cost function $C^*_{u}$ (unknown) which follows the population’s true cost function distribution $D^*$ (unknown). To optimise for EMC, cost function $C_{i}$ are sampled from $D_{train}$, whereas for evaluation the user cost $C^*_{u}$ are simulated from $D_{test}$. We have three plausible distributions which we propose, $D_{mix}$, $D_{perc}$, $D_{lin}$. All three are hierarchical distributions and $D_{mix}$ is a combination of $D_{perc}$, $D_{lin}$. In various experiments we specify the form $D_{train}$ and $D_{test}$ to take the form of one of the proposed distributions $D_{mix}$, $D_{perc}$, $D_{lin}$.


---
&nbsp;
### **General Response**

We thank the reviewers for their comments and time investment in reviewing our work.

Here, we clarify some of the common points. We have updated the paper to improve clarity, please refer to the comment detailing changes made to the paper and the improved notation. We are looking forward to the discussion!


1. **Is our evaluation procedure realistic or preferable to past evaluations?**

In this work, we propose a new evaluation framework where the individuality of users is taken into consideration to evaluate their satisfaction. Each user’s incurred cost and satisfaction is dependent on their simulated cost function $C^*_{u}$ from a test time distribution, $D_{test}$, which is designed to capture how diverse users’ in real scenarios might think of cost functions. $D_{test}$ captures all possible feature subsets which users might consider preferable along with all combinations of linear/percentile costs and provides higher cost to more drastic changes (see section 4.2 and Appendix A.2.1).

2. **Is the evaluation of COLS fair, given that it uses the same distribution at train and test time?**

If our evaluation procedure is sound, we do not see any issue with claiming that COLS does well as a recourse algorithm if it performs well under this evaluation, regardless of the details of the implementation. By rough analogy, our setup is like a supervised learning problem where one uses data from the same distribution, where the test set is not seen during training time. We use cost functions from the same distribution, but the test time cost functions are not seen during recourse generation time. In comparison to past work, this is already a stronger evaluation framework, as several past cost-based methods use the exact same cost function for both recourse generation and evaluation. That all said, we do provide several experiments demonstrating the robustness of our method to distribution shifts between $D_{train}$ and $D_{test}$ in the next question.

3. **What happens if the train and test time cost function distributions differ?**

In Section 4.2, we describe the distribution  $D_{mix}$, $D_{perc}$, and $D_{lin}$ which model how users might think of the cost of transitions. Note that $D_{perc}$, and $D_{lin}$ are two completely different distributions. The first experiment tries to quantify the performance of our method when the train distribution $D_{train}$ is percentile and $D_{test}$ is linear cost based. In Figure 3, the top-left corner corresponds to this case described above, the user satisfaction for the top left corner is similar to the bottom left corner which is $D_{train} = D_{lin}$ and $D_{test} = D_{lin}$. Similarly things happen for the opposite case which is denoted by the top-right ($D_{train} = D_{perc}$, $D_{test} = D_{perc}$) and bottom-right ($D_{train} = D_{lin}$, $D_{test} = D_{perc}$) corners. This means that even if the distributions are different the performance is retained. As mentioned in Appendix B.2, Q5, this is expected when the $D_{train}$ is flexible. In addition, we go on and interpolate the $D_{train}$ and $D_{test}$ from linear to percentile which comprises a grid and we observe that user satisfaction is similar in all settings. Additionally, In Appendix B.2, Q4 Figure 2 we show another type of distribution shift where we observe similar results.

From these experiments, we conclude that COLS is highly robust to some of the most sensible kinds of distribution shifts, which we believe means we could fairly say that the user cost distribution does not actually have to be “known” in advance.

---

### Decision · Program_Chairs · 2022-01-20

**Decision:**

Reject

**Comment:**

This paper makes an interesting contribution to the literature on algorithmic recourse. More specifically, while existing literature assumes that there is a global cost function that is applicable to all the users, this work addresses this limitation and models user specific cost functions. While the premise of this paper is interesting and novel, there are several concerns raised by the reviewers in their reviews and during the discussion: 1) While the authors allow flexibility to model user specific cost functions, they still make assumptions about the kind of cost functions. E.g., they consider three hierarchical cost sampling distributions, each of which model percentile shift, linear shift, and a mixture of these two shifts. The authors do not clearly justify why these shifts and a mixture of these shifts is reasonable. Prior work already considers lot more flexible ways of modeling cost functions (in a global fashion). For example, Rawal et. al. 2020 actually learns costs by asking users for pairwise feature comparisons. Isn't this kind of modeling allowing more flexibility than sticking to percentile/linear shifts and their mixture? 2) Several reviewers pointed out that the main paper does not clearly explain all the key contributions. While the authors have updated their draft to address some part of this concern, reviewers opine that the methods section of the paper still does not discuss the approach and the motivation for the various design choices (e.g., why a mixture of percentile and linear shifts?) clearly. 3) Reviewers also opine that some of the evaluation metrics also need more justification. For instance, Why is fraction satisfied measured at k = 1 i.e, FS@1 measured and why not FS@2 or FS@3? Will the results look different for other values of k here? 4) Given that Rawal et. al. 2020 is a close predecessor of this work, it would be important to compare with that baseline to demonstrate the efficacy of the proposed approach. This comparison is missing.

Given all the above, we are unable to recommend acceptance at this time. We hope the authors find the reviewer feedback useful.